# Provable Posterior Sampling with Denoising Oracles via Tilted Transport

**Joan Bruna**
New York University & Flatiron Institute
bruna@cims.nyu.edu

**Jiequn Han**
Flatiron Institute
jhan@simonsfoundation.org

## Abstract

Score-based diffusion models have significantly advanced high-dimensional data generation across various domains, by learning a denoising oracle (or score) from datasets. From a Bayesian perspective, they offer a realistic modeling of data priors and facilitate solving inverse problems through posterior sampling. Although many heuristic methods have been developed recently for this purpose, they lack the quantitative guarantees needed in many scientific applications. This work addresses the topic from two perspectives. We first present a hardness result indicating that a generic method leveraging the prior denoising oracle for posterior sampling becomes infeasible as soon as the measurement operator is mildly ill-conditioned. We next develop the *tilted transport* technique, which leverages the quadratic structure of the log-likelihood in linear inverse problems in combination with the prior denoising oracle to exactly transform the original posterior sampling problem into a new one that is provably easier to sample from. We quantify the conditions under which the boosted posterior is strongly log-concave, highlighting how task difficulty depends on the condition number of the measurement matrix and the signal-to-noise ratio. The resulting general scheme is shown to match the best-known sampling methods for Ising models, and is further validated on high-dimensional Gaussian mixture models.

## 1 Introduction

Inverse problems consist in reconstructing a signal of interest from noisy measurements. As such, they are a central object of study across many scientific domains, including signal processing, imaging, astrophysics or computational biology. In the common settings where the measurement information is limited, a reliable solution for these problems usually depends on prior knowledge of the data. One popular approach is to choose a regularizer that utilizes data properties such as smoothness or sparseness, and then solve a regularized optimization problem to obtain *a point estimate* of the original data. However, this approach often struggles with selecting an appropriate regularizer and might be unstable in the presence of large measurement noise. A more robust approach takes a statistical formulation and seeks to sample the *posterior distribution* of data based on Bayes's theorem, which allows for uncertainty quantification in the reconstructed data by leveraging a model for the prior data distribution.

While accurate models for high-dimensional distributions are notoriously complex to estimate, the resurgence of deep neural networks has provided unprecedented capabilities for modeling complex data distributions in certain high-dimensional regimes. Specifically, score-based diffusion models [55, 33, 58] have achieved remarkable empirical success in generating high-dimensional data across various domains, including images, video, text, and audio. These models implicitly parameterize data distributions through an iterative denoising process that builds up data from noise. Furthermore, there is a growing literature developing theoretical foundations of score-based diffusion models

38th Conference on Neural Information Processing Systems (NeurIPS 2024).

[17, 7, 45, 16, 19], giving a comprehensive error analysis including score estimation, initialization error and time-discretization error. By generating high-fidelity data, these models can also serve as data prior for posterior sampling in inverse problems in high dimensions. Following this idea, many studies (see, e.g., [40, 22]) have leveraged diffusion models for posterior sampling. However, as discussed below, various categories of approaches for posterior sampling introduce different uncontrollable errors, such as those arising from the approximation of the conditional score or the use of a limited variational family. This abundance of heuristics contrasts with the principled sampling used in prior data generation, and is often at odds with the statistical guarantees needed in many scientific applications.

In this work, we aim to bridge the gap between principled diffusion-based algorithms for both prior and posterior distributions. Focusing on the canonical setting of linear inverse problems, where measurements are of the form $y = Ax + w$, with $x \sim \pi$ the signal to be estimated and $w$ an independent noise, we first illustrate a negative result, revealing that no method can efficiently sample the posterior in general cases, even with the prior denoising oracle. Subsequently, we develop the *tilted transport* technique, which utilizes the quadratic structure of the log-likelihood in linear inverse problems in combination with the prior denoising oracle to exactly transform the original posterior sampling problem into a new one that is easier to sample. Figure 1 illustrates a schematic plot of the method using two-dimensional Gaussian mixture examples, showing that while the original target posterior problem remains multimodal, the boosted posterior resembles a unimodal distribution.

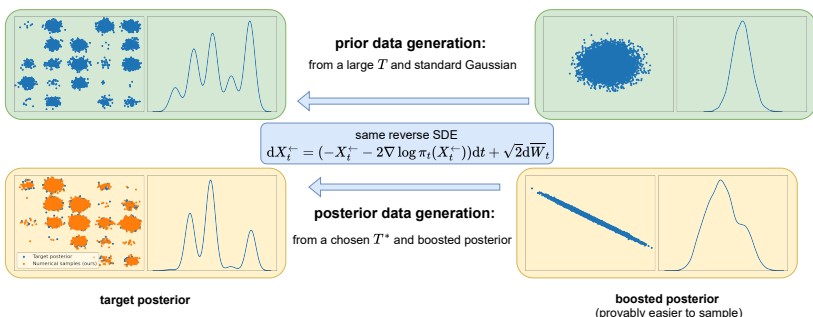

Figure 1: Schematic plot of tilted transport boosting posterior sampling with a 2D Gaussian mixture example. The density plot shows the first variable's density, and the scatter plot displays the samples.

We establish a precise condition where the density of the transformed posterior problem becomes strongly log-concave, making it suitable for efficient sampling via Langevin dynamics. This condition showcases the interplay between a geometric property of the prior (what we call *tilted spread*; see Section 5) and the conditioning and noise level of the measurements. Interestingly, the condition can be satisfied when the signal-to-noise ratio (SNR) is either moderately low or moderately high, in contrast with traditional sampling methods, which typically excel only within a specific regime.

As a first application, we show that tilted transport can sample from Ising models of the form $\nu(x) \propto e^{-\frac{1}{2}x^\top Q x}$, where $x \in \{\pm 1\}^d$ is supported in the hypercube, up to the critical threshold determined by the gap $\lambda_{\max}(Q) - \lambda_{\min}(Q) = 1$, thus matching the performance of Glauber dynamics [24, 1] as well as the computational threshold predicted by the low-degree method [42]. More generally, even when the boosted posterior is not strongly log-concave, it remains easier to sample than the original one. Thus, tilted transport can be combined with any existing black-box posterior sampling methods to enhance their performance. This technique operates without any additional computational cost and functions in a plug-and-play fashion, allowing for straightforward integration into various frameworks. When working with high-dimensional Gaussian mixtures, where an analytical solution to the posterior is available, we numerically validate our theory and demonstrate enhanced posterior sampling performance.

## 1.1 Related Work

Numerous studies in recent years have explored score-based priors for posterior sampling. We note that several recent works [60, 20, 29, 54, 21] introduce hyperparameters to balance the influence of the prior and measurements, resulting in sampling strategies that guide output to regions where the given

observation is more likely. These strategies typically deviate from the principles of Bayesian posterior sampling and often lack a precise definition of the resulting distribution. In contrast, other approaches adhere more closely to Bayesian principles. One such approach is variational inference, which involves designing variational objectives and optimization methods based on the structure of score-based diffusion [41, 47, 26, 37]. However, even with an accurate prior score, the accuracy of posterior sampling heavily depends on the choice of variational family and optimization procedures, not to mention the additional optimization cost. Another popular strategy focuses on approximating the score conditional on the measurement using various heuristics [58, 40, 36, 22, 48, 56, 57]. In this approach, approximation errors typically remain largely uncontrollable due to the challenges associated with tracking the conditional distribution for intermediate states. Recently, some studies have adopted sequential Monte-Carlo methods to systematically approximate the conditional score [62, 13, 23], providing consistency as the number of particles used to approximate the conditional distribution of the intermediate states increases. However, this particle-based method still struggles with high-dimensional problems due to the curse of dimensionality [9]. Alternatively, [61, 63] propose plug-and-play methods with denoising oracles for posterior sampling, offering asymptotic guarantees, though the required steps may grow prohibitively in high dimensions.

We note that [13] also intuitively explores the possibility of reducing the original posterior to an equivalent one under restrictive conditions in the discrete-time setting. In contrast, our tilted transport technique operates in a fairly generic setting and is supported by a clear theoretical foundation. Concurrently, [49] proposes a conceptually similar two-stage approach for posterior sampling in sparse linear regression, based on a different structural prior rather than denoising oracles.

## 2 Preliminaries

**Notations:.** $\mathcal{P}(\mathbb{R}^d)$ denotes the space of probability measures over $\mathbb{R}^d$. $\gamma_d$ denotes the $d$-dimensional standard Gaussian measure, and by slight abuse of notation, $\gamma_\delta$ or $\gamma_\Sigma$ denote the centered Gaussian measure with covariance $\delta I_d$ or $\Sigma$ when the context is clear. For $Q \geq 0$ in $\mathbb{R}^{d \times d}$ and $b \in \mathbb{R}^d$ in the span of $Q$, the quadratic tilt of $\pi$ is the measure $\mathsf{T}_{Q,b}\pi \ll \pi$ with density proportional to $\frac{d\mathsf{T}_{Q,b}\pi}{d\pi}(x) \propto \exp\left\{-\frac{1}{2}x^\top Q x + x^\top b\right\}$. We also use the notation $\mathsf{T}_Q$ when $b = 0$. $\|Q\|$ denotes its operator norm. $\pi \star \gamma$ denotes the convolution of two measures $\pi$ and $\gamma$. For $\alpha \geq 0$ and $\pi \in \mathcal{P}(\mathbb{R}^d)$, we define $\mathsf{D}_\alpha \pi(x) := \alpha^d \pi(\alpha x)$ as the *dilation* of $\pi$. For $\beta \geq 0$ and $\pi \in \mathcal{P}(\mathbb{R}^d)$, we define $\mathsf{C}_\beta \pi(x) := \pi \star (\mathsf{D}_{\beta^{-1/2}}\gamma_d)$ as the *Gaussian convolution* of $\pi$.

**Problem Setup.** Consider a high-dimensional object of interest $x \in \mathbb{R}^d$, drawn from a certain probability distribution $\pi \in \mathcal{P}(\mathbb{R}^d)$. We suppose that one has managed to learn a generative model for $\pi$ via the DDPM objective [33]; in other words, for any $y \in \mathbb{R}^d$ and $\sigma \geq 0$, we have access to the *denoising oracle* $\mathsf{DO}_\pi(y, \sigma) := \mathbb{E}[x|y]$, where $y = x + \sigma w$, with $x \sim \pi$ and $w \sim \gamma_d$ independent. It is now well-established that, such denoising oracle enables efficient sampling of $\pi$, well beyond the classic isoperimetric assumptions for fast relaxation of Langevin dynamics [17].

Suppose that we now measure $y = Ax + \sigma w$, where again $x \sim \pi$ and $w \sim \gamma_{d'}$ are independent, but now $A \in \mathbb{R}^{d' \times d}$ is a *known* linear operator different from the identity. Given these linear measurements, we are now interested in the *posterior sampling* of $x$ given $y$. This corresponds to the basic setup of linear inverse problems, encompassing many applications such as image inpainting, super-resolution, tomography, or source separation, to name a few. We are interested in the following natural question: can the power of denoising oracles be provably transferred to posterior sampling?

By Bayes' rule, the posterior distribution $\nu_{y,A}$ (denoted simply by $\nu$ when the context is clear) has density proportional to $\pi(x)p(y|x) \propto \exp\left\{-\frac{1}{2\sigma^2}\|Ax - y\|^2\right\}\pi(x)$, and thus we can write it as a quadratic tilt of $\pi$:

$$\nu = \mathsf{T}_{Q,b}\pi \,,\ \text{with}\ Q = \sigma^{-2}A^\top A \,,\ b = -\sigma^{-2}A^\top y \,.$$

We readily identify certain regimes where sampling from $\nu$ might be easy:

- If $\lambda_{\min}(Q)$ is sufficiently large, $\lambda_{\min}(Q) \gg 1$, then one expects $\nu$ to be strongly log-concave, enabling fast relaxation of Langevin dynamics.

- If $\lambda_{\max}(Q)$ is sufficiently small, $\lambda_{\max}(Q) \ll 1$, then one expects $\nu \approx \pi$ in the appropriate sense, and therefore that samples from $\pi$ (which can be produced efficiently thanks to $\mathsf{DO}_\pi$) may be perturbed into samples from $\nu$.
- If $A \in O_d$ is a unitary transformation, then $Q = \sigma^{-2}\mathrm{Id}$ and the inverse problem reduces to isotropic Gaussian denoising, and is thus at first glance 'compatible' with the structure of the denoising oracle (such observation will be formalized later).

At this stage, we can already identify two key parameters of the problem that are likely to drive the difficulty of posterior sampling: on one hand, a proxy for the signal-to-noise ratio, measured e.g., by $\mathrm{SNR} := \lambda_{\min}(Q) = \frac{\lambda_{\min}(A)^2}{\sigma^2}$. On the other hand, the conditioning of the measurement operator $A$, $\kappa(A) := \frac{\lambda_{\max}(A)}{\lambda_{\min}(A)}$. As we shall see, these two characteristics of the linear measurement system will characterize necessary and sufficient conditions for probable posterior sampling. In the following, we assume the log of prior density $\pi$ is smooth and its Hessian exists $\forall x \in \mathbb{R}^d$.

**Denoising Oracles and Score-Based Diffusion.** Let us first review the natural connection between denoising and score-based generative modeling. Score-based diffusion models consist of two processes: a forward process that gradually adds noise to input data and a reverse process that learns to generate data by iteratively removing this noise. For example, one widely used family for the forward process is the Ornstein–Uhlenbeck (OU) process[1]:

$$\mathrm{d}X_t = -X_t\mathrm{d}t + \sqrt{2}\mathrm{d}W_t, \quad X_0 \sim \pi, \tag{1}$$

where $W_t$ is the standard Wiener process. We use $\pi_t$ to denote the density of $X_t$, given by the action of the OU semigroup $\pi_t = \mathsf{O}_t^*\pi$, defined by $\mathsf{O}_t f(x) = \mathbb{E}[f(X_t)|X_0 = x]$, and explicitly given by dilated Gaussian convolutions, $\mathsf{O}_t^* := \mathsf{C}_{\beta_t}\mathsf{D}_{\alpha_t}$, with $\beta_t = 1 - e^{-2t}$ and $\alpha_t = e^t$. With a sufficiently large $T$, we know that $\pi_T$ is close to the density of standard Gaussian $\gamma_d$, owing to the exponential contraction of the OU semigroup: $\mathrm{KL}(\pi_T||\gamma_d) \le e^{-T}\mathrm{KL}(\pi||\gamma_d)$.

Finally, the measure $\pi_t$ solves the Fokker-Plank equation

$$\partial_t \pi_t = \nabla \cdot (x\pi_t) + \Delta\pi_t, \quad \pi_0 = \pi. \tag{2}$$

By writing (2) as a transport equation $\partial_t \pi_t = \nabla \cdot ((x + \nabla \log \pi_t)\pi_t)$, we can formally reverse the transport starting at a large time $T$ and solving

$$\partial_t \tilde{\pi}_t = \nabla \cdot (-(x + \nabla \log \pi_{T-t})\tilde{\pi}_t), \quad \tilde{\pi}_0 = \pi_T. \tag{3}$$

Since $\tilde{\pi}_t = \pi_{T-t}$ for $0 \le t \le T$, introducing again the dissipative term leads to $\partial_t \tilde{\pi}_t = \nabla \cdot (-(x + 2\nabla \log \tilde{\pi}_t)\tilde{\pi}_t) + \Delta\tilde{\pi}_t$, $\tilde{\pi}_0 = \pi_T$, which admits the SDE representation

$$\mathrm{d}\tilde{X}_t = (\tilde{X}_t + 2\nabla \log \pi_{T-t}(\tilde{X}_t))\mathrm{d}t + \sqrt{2}\mathrm{d}\overline{W}_t, \quad \tilde{X}_0 \sim \pi_T. \tag{4}$$

In practice, one runs this reverse diffusion starting from $\tilde{X}_0 \sim \gamma_d$ rather than $\tilde{X}_0 \sim \pi_T$. However, by the data-processing inequality, we have that $\mathrm{KL}(\pi||\tilde{\pi}_T) \le \mathrm{KL}(\pi_T||\gamma_d) = O(e^{-T})$, thus incurring in insignificant error. To facilitate later exposition, we write the above process reverse in time [2, 31]

$$\mathrm{d}X_t^\leftarrow = (-X_t^\leftarrow - 2\nabla \log \pi_t(X_t^\leftarrow))\mathrm{d}t + \sqrt{2}\mathrm{d}\overline{W}_t, \quad X_T^\leftarrow \sim \gamma_d, \tag{5}$$

and interpret the data generation process as running the reverse SDE from $T$ back to 0.

By the well-known Tweedie's formula, and up to time reparametrisation, the denoising oracle is equivalent to the time-dependent score $\nabla \log \pi_t$:

**Fact 1** (Tweedie's formula, [32]). *We have* $\nabla \log \pi_t(x) = -(1 - e^{-2t})^{-1}(x - e^{-2t}\mathsf{DO}_\pi(x, 1 - e^{2t}))$.

**Log-Sobolev Inequality and Fast Relaxation of Langevin Dynamics.** Given a Gibbs distribution $\pi \in \mathcal{P}(\mathbb{R}^d)$ of the form $\pi \propto e^{-f}$, a powerful and versatile method to sample from $\pi$ is to consider the Langevin dynamics

$$\mathrm{d}X_t = -\nabla f(X_t)\mathrm{d}t + \sqrt{2}\mathrm{d}W_t, \quad X_0 \sim \mu_0, \tag{6}$$

---

[1]In practice, it is also common to introduce a positive smooth function $\beta \colon \mathbb{R}_+ \to \mathbb{R}_+$ and consider the time-rescaled OU process $\mathrm{d}X_t = -\beta(t)X_t\mathrm{d}t + \sqrt{2\beta(t)}\mathrm{d}W_t$. Our results could be applied directly to these variants by rescaling time. For the ease of notation, we keep $\beta(t) \equiv 1$ in the main text.

where $\mu_0$ is an arbitrary initial distribution. It is easy to verify that these dynamics define a Markov process that admits $\pi$ as its unique invariant measure. Perhaps less obvious is the fact that the Fokker-Plank equation associated with eq. (6), given by $\partial_t \mu = \nabla \cdot (\nabla f \mu) + \Delta \mu$ (and where $\mu_t$ is the law of $X_t$) is in fact a Wasserstein gradient flow for the relative entropy functional $\mathrm{KL}(\mu||\pi)$ [38]. Under this interpretation, one can quantify the convergence of Langevin dynamics to their invariant measure, i.e., its time to relaxation, by establishing a sharpness or *Polyak-Lowacjevitz* (PL)-type inequality. Indeed, by noticing that $\frac{d}{dt}\mathrm{KL}(\mu||\pi) = -\mathrm{I}(\mu||\pi)$, where $\mathrm{I}(\mu||\pi) = \mathbb{E}_\mu[||\nabla \log \mu - \nabla \log \pi||^2]$ is the Fisher divergence, the PL-type inequality in this setting is given by the *Logarithmic Sobolev Inequality* (LSI): we say that a measure $\pi$ satisfies $\mathrm{LSI}(\rho)$ if for any $\mu \in \mathcal{P}(\mathbb{R}^d)$ it holds $\mathrm{KL}(\mu||\pi) \le \frac{1}{2\rho}\mathrm{I}(\mu||\pi)$ .

This functional inequality directly implies $\mathrm{KL}(\mu_t||\pi) \le e^{-2\rho t}\mathrm{KL}(\mu_0||\pi)$. While for general $\pi$ it is typically hard to establish the LSI, there are two important sources of structure that lead to quantitative (i.e., $\rho = \Omega_d(1)$) bounds: when $\pi$ is a product measure $\pi = \tilde{\pi}^{\otimes d}$ (in which case $\pi$ satisfies LSI with the same constant as $\tilde{\pi}$), and when $\pi$ is strongly log-concave[2], i.e., $-\nabla^2 \log \pi(x) \succeq \alpha I$ for all $x$, in which case the celebrated Bakry-Emery criterion [3] states that $\rho \ge \alpha$.

# 3 Evidence of Computational Hardness in the Generic Case

We start our analysis of posterior sampling by discussing negative results for the general case. Recently, [30] established computational lower bounds for this task using cryptographic hardness assumptions. In this section, we complement these results by illustrating a correspondence with sampling problems on Ising models, leading to an arguably simpler conclusion.

For this purpose, consider $\bar{\pi} = \mathrm{Unif}(\{\pm 1\}^d)$ the uniform measure of the hypercube. Quadratic tilts of $\bar{\pi}$ define generic Ising models, a rich and intricate class of high-dimensional distributions. Since $\bar{\pi}$ is a product measure, its associated denoising oracle becomes a separable function that can be computed in closed-form:

**Fact 2** (Denoising Oracle for $\bar{\pi}$). *Let* $\gamma(t; \mu, \sigma) = \exp\left\{-\frac{1}{2\sigma^2}(t-\mu)^2\right\}$. *Then we have*

$$\mathsf{DO}_{\bar{\pi}}(y, \sigma) = (\phi(y_i; \sigma))_{i=1\ldots d} \ , \ with \ \ \phi(t, \sigma) = \frac{\gamma(t, +1, \sigma) - \gamma(t, -1, \sigma)}{\gamma(t, +1, \sigma) + \gamma(t, -1, \sigma)} \ . \tag{7}$$

Given a symmetric matrix $Q \in \mathbb{R}^{d \times d}$, an Ising model is given by the tilt $\mathsf{T}_Q\bar{\pi} \in \mathcal{P}(\{\pm 1\}^d)$. In our setting, we can thus view such models as the posterior distribution of a linear inverse problem associated with the uniform prior $\bar{\pi}$. Efficiently sampling from Ising models is a fundamental question at the interface of statistical physics and high-dimensional probability, and several works provide evidence of computational hardness under a variety of settings.

Notably, by treating $Q$ as the adjacency matrix of a regular graph, [27] establishes that sampling from $\nu$ is impossible for $\lambda_{\max}(Q) - \lambda_{\min}(Q) \ge 2 + \varepsilon$, for any $\varepsilon > 0$, unless $\mathsf{NP} = \mathsf{RP}$. In other words, for poorly conditioned tilt $Q$ (in the sense that there is a large gap between the smallest and largest eigenvalue), there is no efficient posterior sampling algorithm, *even with the knowledge of the prior denoising oracle*. The threshold can even be reduced to $1 + \varepsilon$ by using a weaker notion of computational hardness [42], given by the *low-degree polynomial method* [4, 43]. Remarkably, this threshold agrees with the current best-known algorithmic results for sampling generic Ising models with Glauber dynamics [25, 1]. Finally, we also mention that when $Q$ is a random Gaussian symmetric matrix, the associated so-called Sherrington-Kirkpatrick (SK) model, has been analyzed with dedicated algorithms. In this setting, it is also known [24] that 'stable' sampling algorithms fail to sample from the SK model as soon as $\lambda_{\max}(Q) - \lambda_{\min}(Q) > 1$. In summary, we have:

**Theorem 3** (Computational Hardness of Sampling Ising Models, [42, 27]). *There exist no general-purpose, efficient posterior sampling algorithms, for $Q$ sufficiently ill-conditioned, even under the knowledge of the prior denoising oracle.*

One could wonder whether this computational hardness comes from the discrete nature of the hypercube. It is not hard to observe that this is not the case: the following proposition, proved in Appendix A, shows a simple reduction from a model where the prior $\bar{\pi}$ is replaced by a smooth mixture of Gaussians $\pi$ centered at the corners of the hypercube, with variance $\delta$.

---

[2]or a suitable perturbation of it via the Hooley-Strook perturbation principle [35]

**Proposition 4** (Hardness extends to smooth priors). *Assume a posterior sampler exists for the smooth prior with TV error $\epsilon$ and $\delta = o(d^{-1/2})$. Then there exists a sampler for the associated Ising model with TV error $1.1\epsilon$.*

In conclusion, one cannot hope for a generic method that leverages the prior denoising oracle to perform efficient posterior sampling, as soon as $A$ is mildly ill-conditioned. Thus, in order to perform provable posterior sampling, one needs to either (i) constraint the measurements, or (ii) exploit structural properties of the prior measure. In the following, we focus on (i), namely providing guarantees for well-conditioned $A$ that leverage the OU semigroup for generic prior distributions.

## 4 Posterior Sampling via Tilted Transport

We now present a simple method that reduces the original posterior sampling problem to another posterior sampling problem with more benign geometry, by leveraging the shared quadratic structure of the posterior tilt and the OU semigroup. The power of the denoising oracle to perform sampling of the prior $\pi$ comes from its ability to run the transport equation (3) in either direction, and leveraging the fact that sampling from $\pi_T$ is easy. To transfer this power to posterior sampling, we can thus attempt to replicate this scheme: can we implement a transport between the posterior $\nu$ and a terminal measure $\nu_T$ that is easy to sample, that only relies on the pre-trained prior $\mathsf{DO}_\pi$?

**A Motivating Example.** Consider first the denoising setting: $y = x + \sigma w$. According to the forward process, we have $p(X_s|X_0) \overset{d}{=} \mathcal{N}(e^{-s}X_0, (1 - e^{-2s})I_d)$. Introduce $T^* > 0$ and define $\tilde{y} = e^{-T^*}y = e^{-T^*}x + e^{-T^*}\sigma w$ such that $p(\tilde{y}|x) \overset{d}{=} \mathcal{N}(e^{-T^*}x, e^{-2T^*}\sigma^2 I_d)$. We match the variance by letting $e^{-2T^*}\sigma^2 = 1 - e^{-2T^*}$, i.e., $T^* = \frac{1}{2}\log(1+\sigma^2)$, then we have $p(\tilde{y}|x) = p(X_{T^*}|X_0)$, which gives $(x, \tilde{y}) \overset{d}{=} (X_0, X_{T^*})$. Therefore, to perform the posterior sampling $p(x|\tilde{y})$, we only need to do the sampling $p(X_0|X_{T^*})$, which can be achieved through the reverse SDE. Specifically, let $X_{T^*} = e^{-T^*}y$ and run the reverse SDE (5) from $T^*$ to 0, then $X_0$ will be the desired posterior.

**Hamilton-Jacobi Equation and Quadratic Tilts.** If $\pi_t$ solves the Fokker-Plank eq. (2), then one can verify that the time-varying potentials $f_t := \log \pi_t$ solve the viscous Hamilton-Jacobi PDE (HJE)

$$\partial_t f_t = \Delta f_t + \|\nabla f_t\|^2 + x \cdot \nabla f_t + d , \quad f_0 = f . \tag{8}$$

Now, the posterior $\nu = \mathsf{T}_{Q,b}\pi$ creates an additional quadratic term in the potential $\log \nu = f - \frac{1}{2}x^\top Q x + x \cdot b$. One could naively hope that this additive quadratic term would still define a solution of the HJE with the tilted initial condition $\tilde{f}_0 = \log \nu$ — or equivalently that the measure $\mathsf{T}_{Q,b}\pi_t$ solves the transport equation (3). Unfortunately, due to the nonlinearity in (8) brought by the terms $\|\nabla f_t\|^2$, this is not the case. However, as we shall see now, this is not far from being true: one just needs to consider *time-varying* quadratic tilts in order to satisfy the HJE.

**Tilt Transport Equation.** We consider then a one-parameter family of distributions $\nu_t$ of the form $\nu_t = \mathsf{T}_{Q_t,b_t}\pi_t$, with $Q_0 = Q$ and $b_0 = b$. As it turns out, one can ensure that $\log \nu_t$ solves the HJE associated with the reverse process by asking that $Q_t, b_t$ satisfy the first-order ODE

$$\begin{cases} \dot{Q}_t = 2(I + Q_t)Q_t , & Q_0 = Q \\ \dot{b}_t = (I + 2Q_t)b_t , & b_0 = b \end{cases} \tag{9}$$

**Theorem 5** (Tilted Transport). *Assume $t < T^*$ such that the ODE (9) is well-defined on $[0, t]$. By initializing $X_t \sim \nu_t$ and run the reverse SDE (5) from $t$ to 0, we have $X_s \sim \nu_s$ for $s \in [0, t]$, specifically, $X_0$ gives the desired posterior.*

**Solution to eq. (9).** Without loss of generality, we assume $d' \leq d$, and the observation operator $A \in \mathbb{R}^{d' \times d}$ has a general singular value decomposition form $A = U\Sigma V^\top$ with non-zero singular values $\lambda_1 \geq \lambda_2 \geq \cdots \geq \lambda_{d'} > 0$. By diagonalizing $Q$ and solving the scalar ODE $\dot{q}_t = 2(1 + q_t)q_t$ for diagonal entries, we have $Q_t = V\text{diag}\left(\frac{e^{2t}}{1+\sigma^2/\lambda_1^2 - e^{2t}}, \cdots, \frac{e^{2t}}{1+\sigma^2/\lambda_{d'}^2 - e^{2t}}, 0, \cdots, 0\right)V^\top$, where the solution is defined up to the blowup time $T^* := \frac{1}{2}\log(1 + \sigma^2/\lambda_1^2) = \frac{1}{2}\log(1 + \lambda_{\max}(Q)^{-1})$. $b_t$ can be further solved from the solution $Q_t$; see Appendix B.2 for more details.

With the explicit solution of $Q_t, b_t$, we can interpret the term $\exp(-\frac{1}{2}x^\top Q_t x + x^\top b_t)$ as the likelihood of the inverse problem with respect to the new prior distribution $\pi_t$ and the corresponding operator. Based on this observation and Theorem 5, we have the following corollary, transforming the original posterior sampling problem to a new posterior sampling problem exactly. We remark that when $A$ is identity, the corollary recovers the analysis we have in the motivating example; see Appendix B.2 for the proof and more discussions.

**Corollary 6.** *Fix $t < T^*$. Sampling from the original posterior is equivalent to a two-step process: first, sample from a new posterior $X_t \sim \nu_t$, and then run the reverse SDE (5) from time $t$ to 0.*

## 5  Quantitative Conditions for Provable Sampling

Now we show that the new posterior sampling problem described above is provably easier to sample than the original posterior sampling problem from two aspects. On the one hand, the (negative) eigenvalues of the quadratic tilt $-\frac{1}{2}x^\top Q_t x + x^\top b_t$ become more negative, essentially meaning that the SNR of the new observation model becomes larger. To be more specific, as $t \to T^*$, $\lambda_{\min}(Q_t) \to \frac{1+\lambda_{\max}(Q)^{-1}}{\lambda_{\min}(Q)^{-1}-\lambda_{\max}(Q)^{-1}} > \lambda_{\min}(Q)$. On the other hand, the new prior distribution $\pi_{T^*}$ becomes closer to a single-mode Gaussian (recall that $\mathrm{KL}(\pi_t || \gamma_d) = O(e^{-t})$), which is also easier to sample. Combining these two arguments, we expect that, as $t$ increases, $\nu_t$ becomes easier to sample due to easier prior and easier likelihood:

$$\nu_t(x) \propto \underbrace{\pi_t(x)}_{\text{easier prior}} \underbrace{\exp\left\{-\frac{1}{2}x^\top Q_t x + x^\top b_t\right\}}_{\text{easier likelihood}}.$$

Let us now quantify the above intuition.

**Sufficient Conditions for Strong Log-Concavity of $\nu_{T^*}$.** We start by giving a simple sufficient condition that ensures that $\nu_{T^*}$ is strongly log-concave. As discussed earlier, this ensures fast relaxation of the Langevin dynamics, enabling efficient sampling from $\nu_{T^*}$ – and therefore of $\nu$ as per Corollary 6. For that purpose, given the prior $\pi \in \mathcal{P}(\mathbb{R}^d)$ and $t \geq 0$, we define

$$\chi_t(\pi) := \sup_{x \in \mathbb{R}^d} \|\mathrm{Cov}[\mathsf{T}_{tI,x}\pi]\|, \tag{10}$$

where the covariance is given by $\mathrm{Cov}[\mu] = \mathbb{E}_{x\sim\mu}[xx^\top] - (\mathbb{E}_{x\sim\mu}[x])(\mathbb{E}_{x\sim\mu}[x])^\top$. $\chi_t(\pi)$ thus measures the largest 'spread' of any tilted measure of the form $\mathsf{T}_{t,x}\pi$. Equipped with this definition, we have the following sufficient condition to ensure that $\nu_{T^*}$ is strongly log-concave:

**Proposition 7** (Strong Log-Concavity of $\nu_{T^*}$). *$\nu_{T^*}$ is strongly log-concave if*

$$\chi_{\|Q\|}(\pi) < \|Q\|^{-1} \frac{\kappa(A)^2}{(\kappa(A)^2 - 1)} . \tag{11}$$

The proof is in Appendix C. It relates two parameters of the measurement process, the condition number of $A$ and the signal-to-noise ratio in terms of $\|Q\|$, with a geometric property of the prior, the spread function $\chi_t(\pi)$. While this function is not immediately transparent, the following examples illuminate its behavior in reprsentative high-dimensional settings.

**Example 8** (Behavior of $\chi_t(\pi)$). *We have the following examples*

(i) *Gaussian measure: $\chi_t(\gamma_d) = \frac{1}{1+t}$.*

(ii) *Compactly Supported Gaussian Mixture: If $\mu$ is compactly-supported in a ball of radius $R$ and $\delta \geq 0$, then $\chi_t(\mu \star \gamma_\delta) \leq \left(\frac{R}{1+\delta t}\right)^2 + \frac{\delta}{1+\delta t}$.*

(iii) *Tensorization: If $\mu = \mu_1 \otimes \mu_2 \cdots \otimes \mu_d$, then $\chi_t(\mu) = \max_i \chi_t(\mu_i)$.*

(iv) *Uniform measure on hypercube: If $\pi$ is uniform on the hypercube $\mathcal{H}_d$, then $\chi_t(\pi) = 1$.*

**Ising Models.** As a direct consequence of Proposition 7 and Example 8 (iv), we establish a sampling guarantee for Ising models:

**Corollary 9** (Tilted Transport for the Ising Model). *Let $\pi$ be the uniform measure on the hypercube, and $Q$ such that $\lambda_{\max}(Q) - \lambda_{\min}(Q) < 1$. Then $\nu_{T^*}$ is strongly log-concave, and therefore $\nu = \mathsf{T}_Q \pi$ can be sampled efficiently (in continuous-time).*

This result thus establishes that Ising models admit an efficient *continuous-time* procedure for sampling provided their spectrum satisfies $\lambda_{\max}(Q) - \lambda_{\min}(Q) < 1$, thus precisely matching the threshold of [25, 1] achieved by Glauber dynamics, as well as the low-degree prediction from [42]. We remark though that our procedure is not (yet) algorithmic; a careful analysis of the discrete-time complexity and the approximation rates is beyond the current scope, but our next endeavor. If one specializes the previous result to the SK model, the equivalent inverse temperature that guarantees sampling is $\beta^* = 1/4$, which remains below $\beta = 1$, the threshold of the hard phase. For this threshold, dedicated AMP-based sampling succeeds [24, 14]. We also remark that, in itself, it should not come as a surprise that $\nu$ may be sampled under these conditions, since [5] already established an LSI on $\nu$ directly, using an entropy decomposition along the so-called Polchinski renormalization group [6] that refines our Bakry-Emery criterion. In this context, it would be interesting to explore whether this refined criterion could be applied to $\nu_{T^*}$ to improve upon Proposition 7 under appropriate conditions.

**Gaussian Mixtures.** By applying Proposition 7 to Example 8 (ii), we directly obtain the following guarantee for generic comptactly supported Gaussian mixtures:

**Corollary 10** (Tilted Transport for Gaussian Mixtures). *If $\pi = \mu \star \gamma_\delta$ and $\operatorname{diam}(\operatorname{supp}(\mu)) \leq R$, then $\nu_{T^*}$ is strongly log-concave if*

$$\frac{(1 + \delta\mathrm{SNR}^2)(\delta\kappa(A)^2 + \mathrm{SNR}^{-2})}{\kappa(A)^2 - 1} > R^2 . \tag{12}$$

*It also holds when $\delta = 0$ and the prior $\pi$ is any distribution with a bounded support radius $R$.*

Figure 2 displays several contours of the condition in eq. (12) as a function of SNR and $\kappa(A)$. Each $U$-shaped contour is determined by a combination of $\delta$ and $R$, which uniquely characterizes the prior. For all points $((\mathrm{SNR}), \kappa(A))$ outside of a contour, representing a specific inverse problem, $\nu_{T^*}$ is strongly log-concave and thus easy to sample. Given an observation model where both SNR and $\kappa(A)$ are fixed, it is straightforward to see that the condition in eq. (12) is more readily satisfied as $\delta$ increases and $R$ decreases. Figure 2 also confirms this result since as $\delta$ increases or $R$ decreases, the $U$-shaped contour shrinks and the region of easy to sample expands. Now we discuss the implications in the reverse scenario where the prior is fixed and the observation model is adjusted. If we look at Figure 2 horizontally, we know that given a prior and $\kappa(A)$, the target posterior can be reliably sampled if the SNR is either sufficiently low or high, with the region of mid-SNR being challenging. The closer $\kappa(A)$ is to 1, the smaller this challenging region is. When the problem is denoising such that $\kappa(A) = 1$, the challenging region vanishes, and sampling the posterior is straightforward using the denoising oracle, as previously explained.

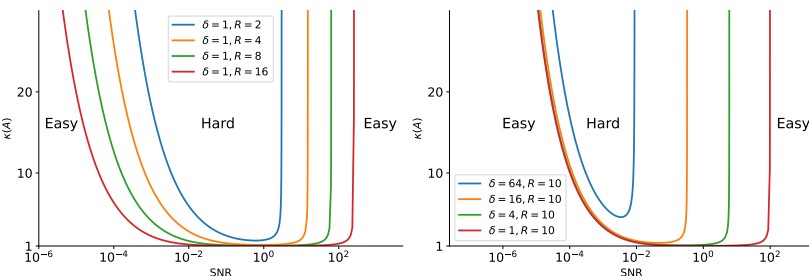

Figure 2: Phase diagram for the boosted posterior $\nu_{T^*}$ being strongly log-concave in Corollary 10.

**Comparisons.** 1. (with Langevin dynamics) As introduced above, Langevin dynamics and its discretized version, Langevin Monte Carlo (LMC) [52, 46] serve as ideal baselines for efficient posterior sampling in high SNR regimes where the posterior becomes strongly log-concave. Proposition 7

demonstrates that our tilted transport technique enables provably efficient sampling from a broader range of prior distributions compared to traditional Langevin dynamics without a denoising oracle. Particularly, in low SNR regimes where conventional Langevin dynamics struggle with severe non-log-concavity and slow mixing times, tilted transport can transform the sampling challenge into a tractable problem for log-concave distributions.

2. (with Importance Sampling) In the low SNR regime with a well-conditioned $A$, the posterior measure can be viewed as a small perturbation of the prior. As such, a natural baseline for posterior sampling is importance sampling using the prior as a proposal — for which samples can be efficiently obtained thanks to the denoising oracle and the variance of sample weights is small. However, as detailed in Appendix C.2, the sampling complexity is exponential with the SNR when the SNR is sufficiently large, assuring the failure of the importance sampling on this extreme.

**Stability.** In the numerical implementation of the boosted posterior, we often encounter specific errors. Appendix C.3 provides a stability analysis of the two-step process outlined in Corollary 6, focusing on initialization error and score error, and demonstrates that the quality of the final samples is robust with respect to these errors.

## 6 Numerical Experiments

Our theory above demonstrates that $\nu_{T^*}$ is provably easier to sample than the original posterior $\nu$. Thus, given a baseline sampling algorithm $\mathsf{Alg}$, we can first sample from the boosted posterior and then apply the denoising oracle to obtain the final sample, rather than directly sampling from $\nu$ using $\mathsf{Alg}$. Algorithm 8 provides a complete description of this approach using tilted transport. In this instance, we use Euler discretization with equal time steps to transport samples from from $\mathsf{T}_{Q_{\tilde{T}},b_{\tilde{T}}}\pi_{\tilde{T}}$ to $\mathsf{T}_{Q,b}\pi$, though alternative time integrators and grids can also be applied.

---

**Algorithm 1** Sampling Using Tilted Transport

---

**Require:** Parameters of quadratic tilt $Q, b$, small time shift $\epsilon$, baseline sampling algorithm $\mathsf{Alg}$, time-dependent score $\nabla \log \pi_t(\cdot)$, $\Delta t$ for reverse SDE
**Ensure:** A sample $X_0$ from posterior distribution $\mathsf{T}_{Q,b}\pi$
 1: Calculate the blowup time by $T^* := \frac{1}{2}\log(1 + \lambda_{\max}(Q)^{-1})$
 2: Determine the number of reverse SDE steps by $N = \lceil \frac{T^*-\epsilon}{\Delta t} \rceil$ and starting time $\tilde{T} = N\Delta t$
 3: Use baseline sampling algorithm $\mathsf{Alg}$ to sample $X_N$ from $\mathsf{T}_{Q_{\tilde{T}},b_{\tilde{T}}}\pi_{\tilde{T}}$
 4: **for** $i = N$ **to** 1 **do**
 5:     Sample $Z_i \sim \mathcal{N}(0, I_d)$
 6:     $X_{i-1} \leftarrow X_i + (X_i + 2\nabla \log \pi_{i\Delta t}(X_i))\Delta t + \sqrt{2\Delta t}\, Z_i$
 7: **end for**
 8: **return** $X_0$

---

We now validate our theoretical results by applying Algorithm 8 to the Gaussian mixture model in high dimensions, using LMC as the baseline algorithm. Same to the models considered in [13], the prior distribution is a mixture of 25 components with known means and variances (see Figure 1 for a 2D visualization and Appendix E for detailed settings). We examine three cases where $d = 20, 40$, and 80. In each scenario, we set $d' = d$, fix $\kappa = 20$, and vary the SNR from $10^{-5}$ to $10^{-1}$. We use the Sliced Wasserstein distance as a principled error metric, computed from samples obtained by our algorithms and samples directly from the analytically computed posterior Gaussian mixture. Figure 3 illustrates the comparison between LMC and LMC boosted by tilted transport. As analyzed earlier, LMC is effective when the SNR is high enough to render the target posterior strongly log-concave, but its error quickly increases as the SNR decreases. In contrast, the tilted transport enhances LMC to perform well in both low and high SNR regimes with small sampling errors. Its performance is weaker in the mid-SNR regime compared to the extremes, as predicted by Corollary 10. However, the tilted transport still improves upon LMC in this challenging regime by boosting effective SNR and simplifying the prior.

We further test tilted transport when $d' < d$, in which $\lambda_{\min}(Q_t)$ remains zero but the signal corresponding to the non-zero eigenvalues still gets enhanced. Therefore, although it becomes more

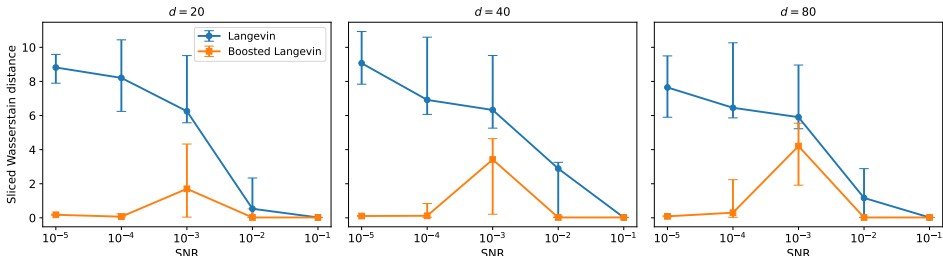

Figure 3: Comparison of Langevin and boosted Langevin for Gaussian mixture prior. We generate the prior, measurement and sample the posterior under 20 different instances in each setting. The sliced Wasserstein distances are plotted with the median in the middle, and the 25th and 75th percentiles indicated by the error bars.

difficult for $\nu_{T^*}$ to be strongly log-concave, the tilted transport can still make the new posterior easier to sample even if it is not strongly log-concave yet. Detailed results are reported in Appendix E.1.

## 7 Discussion and Future Work

In this paper, we theoretically investigate posterior sampling using powerful priors provided by denoising oracles. We demonstrate that efficient posterior sampling can be challenging even with a perfect denoising oracle for the prior. To achieve provable posterior sampling, one must either constrain the measurements or leverage the structural properties of the prior. We focus on the former, showing that well-conditioned measurements enable the proposed tilted transport technique to simplify the task significantly, providing a clear, verifiable condition for efficient sampling, as demonstrated on the Ising model. Several questions remain open: Can this approach provably handle poorly-conditioned measurements, such as inpainting? Can it be extended from linear to nonlinear inverse problems? We show in Appendix D how to extend the tilted transport beyond the condition of Proposition 7 via 'iterated tilts', at the expense of introducing approximation errors. On the theory side, the key object underlying the success of the tilted transport is the spread $\chi_t(\pi)$; in particular, understanding when one can remove dimension dependence is an interesting question. We also aim to systematically evaluate the empirical performance of tilted transport in imaging and scientific computing. Appendix E.2 provides a proof-of-concept for various imaging tasks. We suspect that tilted transport could even improve existing posterior point estimate methods by boosting SNR and enabling proper uncertainty quantification through the reverse process.

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

# A  Proof of Proposition 4

Let $\delta > 0$ and $\bar{\pi}$ be the uniform measure in the $d$-dimensional hypercube. Consider a Gaussian mixture prior $\pi$ defined as $\pi = \bar{\pi} \star \gamma_\delta$.

Since both $\bar{\pi}$ and $\gamma_\delta$ are product measures, it follows that $\pi$ is also a product measure, and therefore its denoising oracle $\mathrm{DO}_\pi$ is explicitly given by $\mathrm{DO}_\pi(y, t)_i = \psi(y_i, t)$, with

$$\psi(v, t) = \int_{\mathbb{R}} u q_{v,t}(u) \mathrm{d}u , \tag{13}$$

$$q_{v,t}(u) = Z^{-1} \left( e^{-\frac{1}{2}(\delta^{-2}(u-1)^2 + t^{-2}(v-u)^2)} + e^{-\frac{1}{2}(\delta^{-2}(u+1)^2 + t^{-2}(v-u)^2)} \right) . \tag{14}$$

Observe that $q_{v,t}$ is the density of a Gaussian mixture in $\mathbb{R}$ of the form $\alpha \mathcal{N}(b_-, \sigma) + (1-\alpha)\mathcal{N}(b_+, \sigma)$, with parameters

$$\sigma^{-2} = \delta^{-2} + t^{-2} \tag{15}$$

$$b_\pm = \frac{\pm\delta^{-2} + t^{-2}v}{\sigma^{-2}} \tag{16}$$

$$\alpha = \frac{e^{\frac{(\delta^{-2}+t^{-2}v)^2}{2\sigma^2}}}{e^{\frac{(\delta^{-2}+t^{-2}v)^2}{2\sigma^2}} + e^{\frac{(-\delta^{-2}+t^{-2}v)^2}{2\sigma^2}}} , \tag{17}$$

and thus $\psi(v, t) = \alpha b_- + (1 - \alpha)b_+$.

Let us now denote by $\mu_Q$ the target Ising model, supported in the $d$-dimensional hypercube, and define the approximation $\mu_Q^\sigma := \mathsf{T}_Q \pi$. Suppose that there is an algorithm $\mathcal{A}$ that leverages the denoising oracle of $\pi$ that can efficiently sample from $\mu_Q^\sigma$: its law $\hat{\mu}$ satisfies $\mathrm{TV}(\mu_Q^\sigma, \hat{\mu}) \leq \epsilon$ with runtime polynomial in $d$ and $\log(\epsilon^{-1})$.

Let now $R(x) = \mathrm{sign}(x)$, and consider the sampler $R \circ \mathcal{A}$, which is now supported in the hypercube. By the triangle and data-processing inequalities, we directly have

$$\mathrm{TV}(R_\# \hat{\mu}, \mu_Q) \leq \mathrm{TV}(R_\# \hat{\mu}, R_\# \mu_Q^\delta) + \mathrm{TV}(R_\# \mu_Q^\delta, \mu_Q) \tag{18}$$

$$\leq \mathrm{TV}(\hat{\mu}, \mu_Q^\delta) + \mathrm{TV}(R_\# \mu_Q^\delta, \mu_Q) \tag{19}$$

$$\leq \epsilon + \mathrm{TV}(R_\# \mu_Q^\delta, \mu_Q). \tag{20}$$

It remains to bound the second term in the RHS. We have to compare two measures in the hypercube. For $\sigma \in \mathcal{H}_d := \{\pm 1\}^d$, they are given respectively by

$$\mu_Q(\sigma) = \frac{1}{Z} e^{-\frac{1}{2}\sigma^\top Q \sigma} , \tag{21}$$

$$R_\# \mu_Q^\delta(\sigma) = \frac{1}{\tilde{Z}} \sum_{z \in \mathcal{H}_d} \int_{R(x)=\sigma} e^{-\frac{1}{2}(x^\top Q x + \delta^{-2}\|x-z\|^2)} \mathrm{d}x . \tag{22}$$

Applying the Laplace approximation into each integral we obtain, as $\delta \to 0$,

$$\int_{R(x)=\sigma} e^{-\frac{1}{2}(x^\top Q x + \delta^{-2}\|x-z\|^2)} \mathrm{d}x = \begin{cases} \sim C_{d,\delta} e^{-\frac{1}{2}\sigma^\top Q \sigma} & \text{if } z = \sigma , \\ \sim C_{d,\delta} e^{-\frac{1}{2}(\sigma \oplus z)^\top Q(\sigma \oplus z)} e^{-\frac{|\sigma-z|}{2\delta^2}} & \text{otherwise} , \end{cases} \tag{23}$$

where $\sigma \oplus z$ is the XOR, and $|\sigma - z|$ is the Hamming distance. We thus have, for any $\sigma \in \mathcal{H}_d$,

$$\left| \tilde{C} R_\# \mu_Q^\delta(\sigma) - e^{-\frac{1}{2}\sigma^\top Q \sigma} \right| \leq 2^d e^{d\lambda_{\min}(Q)/2} e^{-\frac{1}{2}\delta^{-2}} \tag{24}$$

$$\leq e^{-\frac{1}{2}\sigma^\top Q \sigma} 2^d e^{d(\lambda_{\min}(Q)+\lambda_{\max}(Q))/2} e^{-\frac{1}{2}\delta^{-2}} . \tag{25}$$

It follows that we can write $R_\# \mu_Q^\delta(\sigma)$ as

$$R_\# \mu_Q^\delta(\sigma) = C(e^{-\frac{1}{2}\sigma^\top Q \sigma} + \eta_\sigma) ,$$

with a relative error

$$\frac{|\eta_\sigma|}{e^{-\frac{1}{2}\sigma^\top Q\sigma}} \le e^{d(1+\frac{1}{2}(\lambda_{\min}(Q)+\lambda_{\max}(Q)))-\delta^{-2}/2} := \theta \ . \tag{26}$$

It follows that

$$\mathrm{TV}(R_\#\mu_Q^\delta, \mu_Q) = O(\theta) \ , \tag{27}$$

and thus if $\delta \ll \frac{1}{\sqrt{d}}$, we have a negligible TV approximation.

## B   Proofs of Section 4

### B.1   Proof of Theorem 5

*Proof.* We denote the time-dependent score function $\nabla \log \pi_t(x)$ by $s_t(x)$. As derived in Section 2, if we initialize $X_\tau$ according to density $\rho_\tau$ and run the reverse SDE eq. (5), the density of $X_t$ for $t \le \tau$, denoted by $\rho_t$, satisfies the backward PDE:

$$\partial_t \rho_t = \nabla \cdot ((x + 2s_t)\rho_t) - \Delta\rho_t. \tag{28}$$

We need to verify that $v_t$ satisfies the above PDE. Note that a general positive function $\rho_t$ satisfies this PDE is equivalent to that $h_t = \log \rho_t$ satisfies the following Hamilton-Jacobi PDE

$$\partial_t h_t = d + 2\nabla \cdot s_t + \nabla h_t \cdot (x + 2s_t) - (\Delta h_t + \|\nabla h_t\|^2). \tag{29}$$

By definition, we know $h_t = \log \pi_t$ satisfies the above PDE, and we need to prove $h_t = \log v_t = \log \pi_t - \frac{1}{2}x^\top Q_t x + x^\top b_t + F(t)$ satisfies this PDE as well. Here $F(t)$ denotes the normalizing constant. Taking the difference between two equations, we need

$$-\frac{1}{2}x^\top \dot{Q}_t x + x^\top \dot{b}_t + \dot{F} = (-Q_t x + b_t) \cdot (x + 2s_t) + \mathrm{trace}(Q_t) + \|s_t\|^2 - \|s_t - Q_t x + b_t\|^2 \tag{30}$$

$$\Leftrightarrow -\frac{1}{2}x^\top \dot{Q}_t x + x^\top \dot{b}_t + \dot{F} = x^\top(-Q_t - Q_t^\top Q_t)x + x^\top(b_t + 2Q_t^\top b_t) + \|b_t\|^2 + \mathrm{trace}(Q_t) \tag{31}$$

which can be satisfied by the ODE dynamics (9). ∎

### B.2   Derivation of Solution to eq. (9)

**Sanity Check for the Motivating Example.**   In the denoising setting, we have $Q_0 = \frac{1}{\sigma^2}I_d, b_0 = \frac{1}{\sigma^2}y$. The ODE (9) has the explicit solution

$$Q_t = \frac{e^{2t}}{1 + \sigma^2 - e^{2t}}I_d.$$

Note that this solution has a finite blow-up time when $1 + \sigma^2 - e^{2t} \to 0^+$, which is exactly at $T^* = \frac{1}{2}\log(1 + \sigma^2)$, as derived in the main text by matching the SNR. As $t \to T^*$, $Q_t \to \infty I_d$,

$$v_t = \exp\left(f_t(x) - \frac{1}{2}x^\top Q_t x + x^\top b_t + F(t)\right) \to \mathcal{N}(Q_t^{-1}b_t, Q_t^{-1}).$$

To see the limit of $Q_t^{-1}b_t$, we only need to consider the ODE for each component since $Q$ is diagonal. So we view the above ODE as scalar ODEs. Considering the dynamics of

$$\frac{\mathrm{d}}{\mathrm{d}t}\frac{r}{Q} = \frac{\dot{r}Q - r\dot{Q}}{Q^2} = -\frac{r}{Q}, \tag{32}$$

gives $Q_t^{-1}b_t = e^{-t}Q_0^{-1}b_0$. Therefore

$$\lim_{t \to (T^*)^-} Q_t^{-1}b_t = e^{-T^*}Q_0^{-1}b_0 = e^{-T^*}y,$$

which matches the initial condition derived in the main text for the denoising case. Furthermore, we can explicitly verify that the intermediate distribution of $X_t$ by running the reverse SDE from $v_T^*$ is $v_t$:

$$p(X_t | X_{T^*} = e^{-T^*} y) \propto p(X_t) p(X_{T^*} = e^{-T^*} y | X_t)$$

$$\propto \pi_t(X_t) \exp\left(-\frac{1}{2} \frac{\|e^{-T^*} y - e^{-(T^*-t)} x\|^2}{1 - e^{-2(T^*-t)}}\right)$$

$$= \pi_t(X_t) \exp\left(-\frac{1}{2} \frac{\|e^{-t} y - x\|^2}{e^{2(T^*-t)} - 1}\right) \tag{33}$$

To match the form of $v_t$, we have $Q_t = \frac{1}{e^{2(T^*-t)} - 1} = \frac{e^{2t}}{1 + \sigma^2 - e^{2t}}$, $Q_t^{-1} b_t = e^{-t} y$, which are the solutions of the ODE (9).

**Solution to eq. (9).** We recall that the observation operator $A \in \mathbb{R}^{d' \times d}$ has a general singular value decomposition form $A = U\Sigma V^\top$ with non-zero singular values $\lambda_1 \geq \lambda_2 \geq \cdots \geq \lambda_{d'} > 0$. By definition, we have $Q_0 = V\text{diag}(\lambda_1^2/\sigma^2, \cdots, \lambda_{d'}^2/\sigma^2, 0, \cdots, 0)V^\top$. By left multiplying $V^\top$ and right multiplying $V$ to the first ODE in (9), we can diagonalize it to scalar equations $\dot{q}_t = 2(1 + q_t)q_t$ for each diagonal entry. Solving this ODE gives

$$Q_t = V\text{diag}\left(\frac{e^{2t}}{1 + \sigma^2/\lambda_1^2 - e^{2t}}, \cdots, \frac{e^{2t}}{1 + \sigma^2/\lambda_{d'}^2 - e^{2t}}, 0, \cdots, 0\right) V^\top. \tag{34}$$

Here we explain how to solve $b_t$ from eq. (9). We denote $V = [v_1, \cdots, v_d]$, in which $v_i$ are eigenvectors of $Q$ (and $Q_t$ as well), and denote the eigenvalues of $Q_t$ ($0 \leq t < T^*$) by

$$\tilde{\lambda}_i(t) = \begin{cases} \dfrac{e^{2t}}{1 + \sigma^2/\lambda_i^2 - e^{2t}}, & 1 \leq i \leq d' \\ 0, & d' + 1 \leq i \leq d \end{cases} \tag{35}$$

By definition, we know $\tilde{\lambda}_i$ satisfies the ODE

$$\dot{\tilde{\lambda}} = 2(1 + \tilde{\lambda})\tilde{\lambda}.$$

We rewrite the solution $b_t = \sum_i^d \xi_i(t) v_i$ and aim to solve $\xi_i(t)$. From $b_0 = V(\frac{1}{\sigma^2} \Sigma^\top U^\top y)$, we have the initial condition

$$\xi_i(0) = \begin{cases} \dfrac{\lambda_i}{\sigma^2} (U^\top y)_i, & 1 \leq i \leq d' \\ 0, & d' + 1 \leq i \leq d \end{cases} \tag{36}$$

Taking the inner product between $v_i$ and both sides of the ODE $\dot{r}_t = (I + 2Q_t)b_t$, we have

$$\dot{\xi}_i(t) = (1 + 2\tilde{\lambda}_i(t))\xi_i(t).$$

Therefore, for $d' + 1 \leq i \leq d$, $\xi_i(t) = 0$. For $1 \leq i \leq d'$, same to the derivation in eq. (32), we have

$$\frac{\mathrm{d}}{\mathrm{d}t} \frac{\xi_i}{\tilde{\lambda}_i} = -\frac{\xi_i}{\tilde{\lambda}_i},$$

which gives

$$\frac{\xi_i(t)}{\tilde{\lambda}_i(t)} = e^{-t} \frac{\xi_i(0)}{\tilde{\lambda}_i(0)}, \tag{37}$$

$$\Rightarrow \left(\frac{e^{2t}}{1 + \sigma^2/\lambda_i^2 - e^{2t}}\right)^{-1} \xi_i(t) = e^{-t} \frac{\sigma^2}{\lambda_i^2} \frac{\lambda_i}{\sigma^2} (U^\top y)_i, \tag{38}$$

$$\Rightarrow \xi_i(t) = \frac{e^t}{\lambda_i(1 + \sigma^2/\lambda_i^2 - e^{2t})} (U^\top y)_i. \tag{39}$$

**Proof of Corollary 6.** Given Theorem 5, we only need to prove that sampling from

$$\nu_t = \mathsf{T}_{Q_t,b_t}\pi_t \; \propto \; \pi_t(x)\exp\left(-\frac{1}{2}x^\top Q_t x + x^\top b_t\right)$$

is equivalent to sampling from a new posterior. Taking $\pi_t$ as the corresponding prior, we only need to show that the factor $\exp\left(-\frac{1}{2}x^\top Q_t x + x^\top b_t\right)$ is the likelihood of certain observation model in the form of $\tilde{y} = A_t x + w$ with $w \sim \gamma_d$. To end, we need to ensure

$$\exp\left(-\frac{1}{2}x^\top Q_t x + x^\top b_t\right) \; \propto \; \exp\left(-\frac{1}{2}\|A_t x - \tilde{y}\|^2\right)$$

Choosing $A_t$ in the standard SVD form $A_t = \Sigma_t V^\top$ where the singular values of $A_t$ (the diagonal entries of $\Sigma_t$) are $\frac{e^t}{(1+\sigma^2/\lambda_i^2 - e^{2t})^{1/2}}$ for $1 \le i \le d'$, the quadratic term is matched. Matching the first order term requires $b_t = A^\top \tilde{y} = V\Sigma_t^\top \tilde{y}$. Further matching the coefficients in the basis of $V$ requires that

$$(\Sigma_t^\top \tilde{y})_i = \xi_i(t) = \frac{e^t}{\lambda_i(1 + \sigma^2/\lambda_i^2 - e^{2t})}(U^\top y)_i, \quad 1 \le i \le d'.$$

It is easy to verify that $\tilde{y} = \Sigma_t' U^\top y$ with $\Sigma_t' = \mathrm{diag}\left(\frac{1}{(\sigma^2+\lambda_1^2(1-e^{2t}))^{1/2}}, \cdots, \frac{1}{(\sigma^2+\lambda_{d'}^2(1-e^{2t}))^{1/2}}\right)$ satisfies the above requirement.

**Remark 11.** *Also, one can directly use the backward transport equation (3) to generate samples with the probability flow ODE [58] backward in time*

$$\mathrm{d}X_t^\leftarrow = (-X_t^\leftarrow - \nabla \log \pi_t(X_t^\leftarrow))\mathrm{d}t \tag{40}$$

*from $X_T^\leftarrow \sim \gamma_d$. Combining the fact that $\nu_t$ satisfies the PDE (5) and $\Delta\nu_t = \nabla\cdot(\nabla\nu_t) = \nabla\cdot(s_t - Q_t x + b_t)$, we have that $\nu_t$ also satisfies the transport equation $\partial_t p_t = \nabla\cdot((s_t + (I+Q_t)x - b_t)p_t)$. Therefore, for any $t < T^*$, by initializing $X_t \sim \nu_t$ and run the reverse ODE $\mathrm{d}X_t^\leftarrow = (-(I+Q_t)X_t^\leftarrow - \nabla\log\pi_t(X_t^\leftarrow) + b_t)\mathrm{d}t$ then $X_0^\leftarrow$ also gives the desired posterior. However, we note that unlike the reverse SDE case, the corresponding transport PDE and the vector field in the reverse ODE case are different from those used in the prior data generation. As discussed below, both $Q_t$ and $b_t$ are singular near $T^*$. Therefore running the reverse SDE might be preferrable for better numerical stability.*

## C   Proofs of Section 5

### C.1   Proof of Proposition 7

*Proof of Proposition 7.* By definition, we need to show

$$-\lambda_{\min}(Q_{T^*}) + \sup_x \lambda_{\max}(\nabla^2 \log \pi_{T^*}(x)) < 0. \tag{41}$$

From the argument in the main text, we know $T^* = \frac{1}{2}\log(1 + \lambda_{\max}(Q)^{-1})$, and thus

$$\lambda_{\min}(Q_{T^*}) = \frac{1 + \lambda_{\max}(Q)^{-1}}{\lambda_{\min}(Q)^{-1} - \lambda_{\max}(Q)^{-1}}.$$

From Corollary 13, we have

$$\sup_x \lambda_{\max}(\nabla^2 \log \pi_{T^*}(x)) \le (1 + \|Q\|)\left(\|Q\|\chi_{\|Q\|}(\pi) - 1\right). \tag{42}$$

Let $m = \lambda_{\min}(Q)$. Therefore, we can guarantee that $\nu_{T^*}$ is strongly log-concave if

$$\frac{1 + \|Q\|^{-1}}{m^{-1} - \|Q\|^{-1}} > (1 + \|Q\|)\left(\|Q\|\chi_{\|Q\|}(\pi) - 1\right), \tag{43}$$

or equivalently

$$\chi_{\|Q\|}(\pi) < \|Q\|^{-1}\frac{\kappa^2(A)}{\kappa^2(A) - 1}. \tag{44}$$

■

**Lemma 12** (Hessian of Gaussian Mixture Potential). *Let $\pi = \mu \star \gamma_\Sigma$ be a Gaussian mixture. Then* $\nabla^2 \log \pi(x) = \Sigma^{-1} \left( \text{Cov} \left[ T_{\Sigma^{-1}, \Sigma^{-1}x} \mu \right] \Sigma^{-1} - I \right)$ .

*Proof.* Let us first compute the score $\nabla \log \pi$. By definition we have

$$\nabla \log \pi(x) = -\Sigma^{-1} \left( x - \frac{\int y\mu(y) e^{-\frac{1}{2}(x-y)^\top \Sigma^{-1}(x-y)} dy}{\int \mu(y) e^{-\frac{1}{2}(x-y)^\top \Sigma^{-1}(x-y)} dy} \right) \tag{45}$$

$$= -\Sigma^{-1}(x - \mathbb{E}\left[ T_{\Sigma^{-1}, \Sigma^{-1}x} \mu \right]) , \tag{46}$$

and thus

$$\nabla^2 \log \pi(x) = \Sigma^{-1} \left( \text{Cov} \left[ T_{\Sigma^{-1}, \Sigma^{-1}x} \mu \right] \Sigma^{-1} - I \right) , \tag{47}$$

where we defined $\text{Cov}[\mu] = \mathbb{E}_{x \sim \mu}[xx^\top] - (\mathbb{E}_{x \sim \mu}x)(\mathbb{E}_{x \sim \mu}x)^\top$. ∎

**Corollary 13.** *In particular, we have*

$$\sup_x \lambda_{\max}(\nabla^2 \log \pi_{T^*}(x)) \leq (1 + \|Q\|) \left( \|Q\| \chi_{\|Q\|}(\pi) - 1 \right) . \tag{48}$$

*Proof.* From Lemma 12 and $\pi_t = C_{1-e^{-2t}}(D_{e^t}\pi) = D_{e^t}\pi \star \gamma_{1-e^{-2t}}$, we directly have

$$\nabla^2 \log \pi_t(x) = (1 - e^{-2t})^{-1} \left( (1 - e^{-2t})^{-1} \text{Cov} \left[ T_{(1-e^{-2t})^{-1}, (1-e^{-2t})^{-1}x}(D_{e^t}\pi) \right] - I \right) . \tag{49}$$

Now, using the commutation property between the isotropic tilt and the dilation $D_\alpha T_{\eta,\theta} = T_{\alpha^2\eta, \alpha\theta} D_\alpha$, we have

$$\text{Cov} \left[ T_{(1-e^{-2t})^{-1}, (1-e^{-2t})^{-1}x}(D_{e^t}\pi) \right] = \text{Cov} \left[ D_{e^t} T_{(e^{2t}-1)^{-1}, e^{-t}(1-e^{-2t})^{-1}x}\pi \right] \tag{50}$$

$$= e^{-2t} \text{Cov} \left[ T_{(e^{2t}-1)^{-1}, e^{-t}(1-e^{-2t})^{-1}x}\pi \right] , \tag{51}$$

and therefore

$$\sup_x \left\| \text{Cov} \left[ T_{(1-e^{-2t})^{-1}, (1-e^{-2t})^{-1}x}(D_{e^t}\pi) \right] \right\| \leq e^{-2t} \chi_{(e^{2t}-1)^{-1}}(\pi) . \tag{52}$$

Using that $e^{2T^*} - 1 = \|Q\|^{-1}$, we thus obtain

$$\sup_x \lambda_{\max}(\nabla^2 \log \pi_{T^*}(x)) \leq (1 + \|Q\|) \left( \|Q\| \chi_{\|Q\|}(\pi) - 1 \right) . \tag{53}$$

∎

**Lemma 14** (Isotropic Tilt of a Gaussian Mixture). *If $\pi = \mu \star \gamma_\delta$, then*

$$T_{tI,z}\pi = \tilde{\mu} \star \gamma_{\sigma^2} , \tag{54}$$

*where $\sigma^{-2} = \delta^{-1} + t$ and*

$$\tilde{\mu}(\tilde{y}) \propto \mu((\sigma^{-2}\tilde{y} - z)\delta) e^{\frac{1}{2}(\sigma^{-2}\|\tilde{y}\|^2 - \delta\|\sigma^{-2}\tilde{y} - z\|^2)} . \tag{55}$$

*Proof.* By definition, we have

$$T_{tI,z}\pi \propto \int d\mu(y) e^{-\frac{1}{2}t\|x\|^2 + x \cdot z - \frac{1}{2}\delta^{-1}\|x-y\|^2} .$$

By expressing

$$-\frac{1}{2}t\|x\|^2 + x \cdot z - \frac{1}{2}\delta^{-1}\|x-y\|^2 = -\frac{1}{2}\sigma^{-2}\|x - \tilde{y}\|^2 + C$$

we have

$$\sigma^{-2} = \delta^{-1} + t , \tag{56}$$

$$\tilde{y} = \frac{\delta^{-1}y + z}{\delta^{-1} + t} , \tag{57}$$

$$C = \frac{1}{2} \left[ \sigma^{-2}\|\tilde{y}\|^2 - \delta^{-1}\|y\|^2 \right] , \tag{58}$$

which gives the desired result after performing the affine change of variables from $y$ to $\tilde{y}$. ∎

*Proof of Examples 8.* The first example is immediate, after observing that $\mathsf{T}_t \gamma$ is a Gaussian of variance $(1+t)^{-1}$. For the Gaussian mixture example, we observe from Lemma 14 that $\mathsf{T}_t(\mu \star \gamma_\delta)$ is a Gaussian mixture of variance $(t + \delta^{-1})^{-1}$, where the mixture distribution is supported in a ball of radius $R \frac{\delta^{-1}}{\delta^{-1}+t}$. Moreover, the covariance of a homogeneous mixture of the form $\mu \star \gamma_\Sigma$ is $\Sigma + \mathrm{Cov}(\mu)$.

If $\mu$ is a product measure, we observe that the isotropic tilt $\mathsf{T}_t \mu$ is also a product measure, and therefore its covariance is diagonal. Finally, by the previous argument, if $\pi$ is the uniform measure on the hypercube, then $\chi_t(\pi) = \chi_t(\frac{1}{2}(\delta_{-1} + \delta_{+1})) = 1$. ∎

*Proof of Corollary 10.* We plug the spread function $\chi_t(\pi) = \chi_t(\mu \star \gamma_\delta) \leq \left(\frac{R}{1+\delta t}\right)^2 + \frac{\delta}{1+\delta t}$ from Example 8 (ii) into eq. (12) to get (we use $\kappa$ to denote $\kappa(A)$ for simplicity

$$\|Q\|^{-1} \frac{\kappa^2}{(\kappa^2 - 1)} > \left(\frac{R}{1 + \delta\|Q\|}\right)^2 + \frac{\delta}{1 + \delta\|Q\|} \tag{59}$$

$$\Leftrightarrow (1 + \delta\|Q\|) \left(\frac{((1+\delta\|Q\|))\kappa^2}{\|Q\|(\kappa^2 - 1)} - \delta\right) > R^2 \tag{60}$$

$$\Leftrightarrow \frac{((1 + \delta\|Q\|)(\kappa^2 + \delta\|Q\|)}{\|Q\|(\kappa^2 - 1)} > R^2 \tag{61}$$

$$\Leftrightarrow \frac{(1 + \delta\mathrm{SNR}^2)(\delta\kappa^2 + \mathrm{SNR}^{-2})}{\kappa^2 - 1} > R^2 \tag{62}$$

∎

## C.2 Exponential Complexity of Importance Sampling in High SNR Regime

As shown in the main text, the proposed boosted posterior provably works for both low SNR and high SNR regimes. In this section, we formally argue that the importance sampling method is a nature baseline for posterior sampling with a large noise (low SNR regime) , but can suffer from exponential complexity when the SNR is high.

In order to estimate an integral of a function $f$ with respect to the posterior measure $\nu$:

$$I(f) := \int_{\mathbb{R}^d} f(x) \mathrm{d}\nu(x),$$

the idea of importance sampling is to independently sample $X_1, \ldots, X_n$ from the prior $\pi$ and calculate

$$I_n(f) := \frac{\sum_{i=1}^n f(X_i)\tau(X_i)}{\sum_{i=1}^n \tau(X_i)},$$

where $\tau(x)$ is the observation likelihood $\exp(-\frac{1}{2}x^\top Q x + x^\top r)$. With the Denoising Oracle, we can sample from the prior efficiently. Intuitively, one can think if the posterior and prior are very similar, for example, when $\sigma$ is large such that the ratio $\tau$ is close to 1, $I_n(f)$ computed from prior samples can efficiently approximate $I(f)$. On the contrary, if $\sigma$ is small, $\tau(x)$ can have very large variance and the importance sampling can be inefficient since many prior proposals have very small weights. The work [15, Theorem 1.2] proves that, in a fairly general setting, a sample of size approximately $\exp(\mathrm{KL}(\nu\|\pi))$ is necessary and sufficient for accurate estimation by importance sampling, where $\mathrm{KL}(\nu\|\pi)$ is the Kullback–Leibler divergence of $\pi$ from $\nu$:

$$\mathrm{KL}(\nu\|\pi) = \int_{\mathbb{R}^d} \log\left(\frac{\mathrm{d}\nu}{\mathrm{d}\pi}\right) \mathrm{d}\nu = \int_{\mathbb{R}^d} (-\frac{1}{2}x^\top Q x + x^\top b)\mathrm{d}\nu(x).$$

This result confirms one part of the intuition above: if $\sigma$ is sufficiently small, then the magnitude of $Q$ and $r$ will be sufficiently small, and so is $(\nu\|\pi)$ and the number of samples needed in the importance sampling. Next we show that for a fairly generic prior distribution $\pi$, when the SNR is large, $\mathrm{KL}(\nu\|\pi)$ will be also large such that we need approximately $O(e^{d \cdot \mathrm{SNR}})$ examples to implement importance sampling, which is unachievable.

Without loss of generality, we assume the covariance of the prior $\pi$ is $I_d$.

**Proposition 15** (Importance Sampling Sample Complexity Lower Bound). *Assume $\nabla \log \pi(x)$ is L-Lipschitz:*

$$\|\nabla \log \pi(x) - \nabla \log \pi(z)\| \le L\|x - z\| . \tag{63}$$

*Then, when* $\mathrm{SNR} > L + 2$*, we have*

$$\mathrm{KL}(\nu\|\pi) \ge O(d \cdot \mathrm{SNR}) , \tag{64}$$

*and therefore the sample complexity of IS is exponential in dimension.*

*Proof.* We wish to lower bound $\mathrm{KL}(\nu\|\pi)$ with tools of functional inequalities for the concentration of measure. By the celebrated work [51, 10] that the log-Sobolev inequality implies the Talagrand transport-entropy inequality, we have

$$\mathrm{KL}(\nu\|\pi) \ge \frac{\mathrm{LSI}(\nu)W(\nu,\pi)^2}{2}. \tag{65}$$

Here $W(\nu, \pi)$ denotes the Wassertain distance between $\nu$ and $\pi$:

$$W(\nu,\pi) = \sqrt{\inf_{\gamma \in \Gamma(\nu,\pi)} \|x - y\|^2 \mathrm{d}\gamma(x,y)} ,$$

where $\Gamma(\nu, \pi)$ denotes the set of probability measures on $\mathbb{R}^d \times \mathbb{R}^d$ with marginals $\nu$ and $\pi$.

First by Equation (63), we have

$$\nabla^2 (-\log \nu) \succeq (\mathrm{SNR} - L)\mathrm{Id} , \tag{66}$$

which gives

$$\mathrm{LSI}(\nu) \ge (\mathrm{SNR} - L) , \tag{67}$$

by Bakry-Emery criterion [3]. Furthermore, we have the lower bound for the Wasserstain distance [28]

$$W^2(\nu,\pi) \ge \|\mathrm{mean}(\nu) - \mathrm{mean}(\pi)\|^2 + \mathrm{trace}(\mathrm{Cov}(\nu) + \mathrm{Cov}(\pi) - 2(\mathrm{Cov}(\pi)^{\frac{1}{2}}\mathrm{Cov}(\nu)\mathrm{Cov}(\pi)^{\frac{1}{2}})^{\frac{1}{2}}) \tag{68}$$

$$\ge \mathrm{trace}(\mathrm{Cov}(\nu) + I_d - 2\mathrm{Cov}(\nu)^{\frac{1}{2}}) \tag{69}$$

$$= \mathrm{trace}(\mathrm{Diag}(\mathrm{Cov}(\nu)) + I_d - 2\mathrm{Diag}(\mathrm{Cov}(\nu))^{\frac{1}{2}}) \tag{70}$$

$$= \sum_{i=1}^{d}((1 - \mathrm{Std}(\nu)_i)^2) , \tag{71}$$

where the second-to-last equality comes from the fact that the trace remains unchanged under orthogonal transformation. By (66) and Brascamp-Lieb Inequality (Theorem 16), we have $\mathrm{Std}(\nu)_i \le \sqrt{\frac{1}{\mathrm{SNR}-L}}$. With the condition $\mathrm{SNR} - L > 2$,

$$W^2(\nu,\pi) \ge \left(1 - \sqrt{\frac{1}{2}}\right)^2 d . \tag{72}$$

Collecting eqs. (65), (67) and (72), we obtain our final estimate

$$\mathrm{KL}(\nu\|\pi) \ge O(d \cdot \mathrm{SNR}) . \tag{73}$$

∎

**Theorem 16** (Brascamp-Lieb Inequality, [11]). *If $\pi$ is a strongly-log-concave measure on $\mathbb{R}^d$, i.e., of the form $\pi = e^{-f}$ with $\nabla^2 f(x) \succeq \alpha \mathrm{Id}$ for all $x \in \mathbb{R}^d$, then $\|\mathrm{Cov}(\pi)\| \le \alpha^{-1}$.*

## C.3 Stability Analysis

In the numerical implementation of boosted posterior, we typically encounter certain errors. Especially, we may have imperfect score subject to certain $L^2$ errors, and we may not be able to sample the boosted posterior $\nu_t$ exactly. Suppose that instead of starting from $\nu_t$ at $t$ and run the exact reverse SDE (5), we start from an approximate distribution $q_t \approx \nu_t$ and run the reverse SDE (5) with approximating score $s_\theta(x,t) \approx \nabla \log \pi_t(x)$ where $\theta$ denote the parameters parametrizing the score. Denote the distribution of the final samples by $q_0$, we have the following error estimate

**Proposition 17.** *Suppose $\nu_t, q_t, \nabla \log \pi_t, s_\theta(x,t)$ has enough regularities such that the reverse SDEs exist, if the Novikov's condition $\mathbb{E}\left[\exp(\int_0^t \|\nabla \log \pi_\tau(x) - s_\theta(x,\tau)\|^2 d\tau)\right] < \infty$, then*

$$\mathrm{KL}(\nu\|q_0) \leq \int_0^t \mathbb{E}_{\nu_\tau} \|\nabla \log \pi_\tau(x) - s_\theta(x,\tau)\|^2 d\tau + \mathrm{KL}(\nu_t\|q_t). \tag{74}$$

The above proposition ensures that if both the initialization error and score error (over the posterior paths) are small, then the distribution of our final samples is close to the target posterior. The proof is provided in Appendix C.3. Note that we consider the reverse dynamics in continuous-time without time discretization error. There are various works [44, 45, 17, 8] analyzing the time discretization error and those techniques can be further incorporated into the above error estimate.

The proof is similar to that in [59]. Here we provide the proof in our posterior sampling context for completeness.

*Proof.* Consider the following two reverse dynamics needed for the error estimate: one is based on the exact score and starts from the exact boosted posterior

$$dX_\tau = (-X_\tau - 2\nabla \log \pi_\tau(X_\tau))d\tau + \sqrt{2}dW_\tau, \quad X_t \sim \nu_t, \tag{75}$$

and another one is based on the approximate score and starts from the approximation to the boosted posterior

$$d\tilde{X}_\tau = (-\tilde{X}_\tau - 2s_\theta(\tilde{X}_\tau,\tau))d\tau + \sqrt{2}dW_\tau, \quad \tilde{X}_t \sim q_t, \tag{76}$$

Note that these two dynamics are defined backwardly for $\tau \in [0,t]$ and we drop the superscript $\leftarrow$ for notation simplicity. We denote the path measure of $\{X_\tau\}_{\tau \in [0,t]}$ and $\{\tilde{X}_\tau\}_{\tau \in [0,t]}$ by $\nu$ and $q$, respectively. Then $\nu$ and $q_0$ are the marginal distributions of the two path measures at $t = 0$. By data processing inequality and chain rule of KL divergence, we have

$$\mathrm{KL}(\nu\|q_0) \leq \mathrm{KL}(\nu\|q) \tag{77}$$

$$\leq \mathrm{KL}(\nu_\tau\|q_\tau) + \mathbb{E}_{z \sim \nu_\tau} \mathrm{KL}(\nu(\cdot|X_t = z)\|q(\cdot|\tilde{X}_t = z)) \tag{78}$$

Given the Novikov's condition, we can apply the Girsanov theorem [50] to eq. (75)76 to compute the second term above

$$\mathbb{E}_{z \sim \nu_\tau} \mathrm{KL}(\nu(\cdot|X_t = z)\|q(\cdot|\tilde{X}_t = z)) \tag{79}$$

$$\leq -\mathbb{E}_{\nu}\left[\log \frac{dq}{d\nu}\right] \tag{80}$$

$$= \mathbb{E}_{\nu}\left[2\int_0^t (\nabla \log \pi_\tau(x) - s_\theta(x,\tau))dW_\tau + \int_0^t \|\nabla \log \pi_\tau(x) - s_\theta(x,\tau)\|^2 d\tau\right] \tag{81}$$

$$= \mathbb{E}_{\nu}\left[\int_0^t \|\nabla \log \pi_\tau(x) - s_\theta(x,\tau)\|^2 d\tau\right] \tag{82}$$

$$= \int_0^t \mathbb{E}_{\nu_\tau} \|\nabla \log \pi_\tau(x) - s_\theta(x,\tau)\|^2 d\tau \tag{83}$$

∎

# D   Iterated Tilted Transport

We have shown that posterior sampling of $\nu = \mathsf{T}_Q\pi$ can be reduced to sampling from $\nu_{T^*}$ by running the reverse SDE. While $\nu_{T^*}$ is easy to sample under the conditions presented in Section 5, these may

not be verified in several situations of interest. In this context, a natural question is whether one could still leverage the tilted transport, at the expense of introducing sampling error. This is what we address in this section.

Let $\lambda_1, \ldots \lambda_d$ be the eigenvalues of $Q$. Let us assume for simplicity that $b = 0$ and all eigenvalues have multiplicity 1, so $\lambda_i > \lambda_{i+1}$. We define the events $T_j^*$ for $j = 1 \ldots d$ given by

$$T_j^* := \frac{1}{2} \log(1 + \lambda_j^{-1}) \, . \tag{84}$$

Denote by

$$\bar{\lambda}_j(t) = \begin{cases} \infty & \text{if } t \geq T_j^* \, , \\ \lambda_j(t) & \text{otherwise,} \end{cases}$$

where $\lambda_j(t) = e^{2t}/(1 + \lambda_j^{-1} - e^{2t})$ is the solution to the ODE $\dot{q}_t = 2(1 + q_t)q_t$. By abusing notation, we denote by $\bar{Q}_t, t \geq T^*$ the matrix that shares eigenvectors with $Q$, and with eigenvalues $(\bar{\lambda}_1(t), \ldots, \bar{\lambda}_d(t))$. Denote by $V_k = [v_{d-k} \ldots v_d] \in \mathbb{R}^{d \times k}$ the orthogonal projection onto the last $k$ eigenvectors.

While previously we considered only the transport between $\nu$ and $\nu_1 := \nu_{T_1^*}$, now we can consider the sequence $\nu_k := \mathsf{T}_{\bar{Q}_{T_k^*}} \pi_{T_k^*}$ for $k = 1, \ldots, d$. Observe that $\nu_k$ is a measure supported on a subspace $\Omega_k$ of dimension $d - k$; in other words, where $k$ directions are singular, corresponding to the eigenvectors associated with the $\infty$ eigenvalues of $\bar{Q}_{T_k^*}$, and thus $\Omega_k = \{x \in \mathbb{R}^d; V_k^\top x = \mathbf{y}_k\}$ for some $\mathbf{y}_k \in \mathbb{R}^k$.

Now, let us consider $k^* = \min\{k; \nu_k \text{ is s.l.c.}\}$; that is, the first $k$ such that $\nu_k$ is strongly log-concave, and therefore efficiently sampleable by Langevin dynamics. Under the same assumptions as Corollary 10, and by defining $\kappa_k := \frac{\lambda_k}{\lambda_d}$ as the condition number of the truncated $Q$, we immediately obtain the bound

$$k^* \leq \min\left\{k; \frac{(1 + \delta^2 \mathrm{SNR})(\delta^2 \kappa_k + \mathrm{SNR}^{-1})}{\kappa_k - 1} > R^2\right\} \, . \tag{85}$$

For $k < k^*$, assume first that one had sampling access to $\nu_{k+1}$. Running the reverse tilted transport for time $\eta_k = T_{k+1}^* - T_k^*$ would produce samples from a tilted measure $\tilde{\nu}_k := \mathsf{T}_{\tilde{Q}_k} \pi_{T_k^*}$, where $\tilde{Q}_k$ has eigenvalues $(\psi(\eta_k), \ldots, \psi(\eta_k), \lambda_{k+1}(T_k^*), \ldots, \lambda_d(T_k^*))$, where we defined $\psi(t) := (e^{-2t} - 1)^{-1}$ as the inverse of $\lambda \mapsto \frac{1}{2} \log(1 + \lambda^{-1})$. It is thus a non-singular measure in $\mathbb{R}^d$, capturing the fact that the denoising oracle driving the reverse dynamics is isotropic, and thus oblivious to the existence of the singular support of $\nu_k$.

We thus need a procedure to transform samples from $\tilde{\nu}_k$ to samples of $\nu_k$. The easiest procedure is to simply marginalize the coordinates $(x_1, \ldots, x_k) = V_k^\top x \in \mathbb{R}^k$, ie

$$\bar{\nu}_k(x_{k+1}, \ldots, x_d) = \int_{\mathbb{R}^k} \tilde{\nu}_k(\mathrm{d}x_1, \ldots, \mathrm{d}x_k, x_{k+1}, \ldots, x_d) \in \mathcal{P}(\mathbb{R}^{d-k}) \, ,$$

and then 'lift' this measure in the subspace $\Omega_k$, i.e.,

$$\hat{\nu}_k(\mathbf{x}_k; \mathbf{x}_{-k}) := \delta(\mathbf{x}_k - \mathbf{y}_k)\bar{\nu}_k(\mathbf{x}_{-k}) \, , \tag{86}$$

where we defined $\mathbf{x}_k = (x_1, \ldots, x_k)$ and $\mathbf{x}_{-k} = (x_{k+1}, \ldots, x_d)$.

We can then iteratively run the tilted transport backwards, from $k = k^* - 1$ to $k = 0$, as illustrated in Algorithm 2:

---
**Algorithm 2** Sampling Using Iterated Tilted Transport
---
1: Start by sampling $X_{k^*} \sim \nu_{k^*}$.
2: **for** $k = k^* - 1$ **to** 0 **do**
3:     Run tilted transport starting at $X_{k+1}$ for time $\eta_k$, resulting in $\tilde{X}$.
4:     Set $X_k = (\mathbf{y}_k; \tilde{X}_{-k})$.
5: **end for**
6: **return** $X_0$

---

By the data-processing inequality, the TV error will accumulate linearly at each step. Denoting $\hat{v}$ the law of $X_0$, we have

$$\mathrm{TV}(\hat{v}, v) \leq \sum_{0 < k < k^*} \mathrm{TV}(\hat{v}_k, v_k) \,. \tag{87}$$

This bound can be interpreted as the accumulation of errors arising from conditioning a measure by marginalizing over its first components. To the extent that $\psi(\eta_k)$ is large, these variables are nearly deterministic, so one would expect that marginalization is a good approximation of conditioning. The outstanding question is to understand conditions when this error guarantee can be quantified.

Inspired by [18], a natural extension of this simple iterative procedure is to apply 'thermalization' towards the stationary measure $v_k$ after line 4 of Algorithm 2 above, by running Langevin dynamics in $\Omega_k$ with score $\nabla \log v_k$:

$$\mathrm{d}X_t = \nabla \log v_k(X_t)\mathrm{d}t + \sqrt{2}\mathrm{d}W_t \,, \ X_0 \sim \hat{v}_k \,. \tag{88}$$

The drift of this diffusion is available, since both $Q_{T_k^*}$ and $\nabla \log \pi_{T_k^*}$ are known, so is $\nabla \log v_k$.

Denote by $\check{v}_k$ the law of $X_t$ after time $t = B_k$. While the time to relaxation of such Langevin dynamics is generally not quantitative (otherwise $k^* \leq k$), even a short amount of thermalization is able to improve upon the previous method. Indeed, by the reverse transport inequality [10, Lemma 4.2], a weaker Wasserstein guarantee $W_2(\hat{v}_k, v_k)$ can be 'upgraded' to a TV guarantee of the form $\mathrm{TV}(\check{v}_k, v_k) = O(\sqrt{L_k}W_2(\hat{v}_k, v_k))$ by running Langevin dynamics for time $B_k = \Theta(1/L_k)$, where $L_k = \sup_x \lambda_{\max}(\nabla^2 \log v_k(x)) > 0$ is the largest eigenvalue of $\nabla^2 \log v_k$, which is positive by definition of $k^*$ and $k < k^*$. In summary, the 'thermalized' iterated tilted transport satisfies an error bound of the form

$$\mathrm{TV}(\hat{v}, v) \lesssim \sum_{0 < k < k^*} \sqrt{L_k}W_2(\hat{v}_k, v_k) \,. \tag{89}$$

# E  Experimental Details

## E.1  Gaussian mixture models

For a given dimension $d$ with $d \bmod 2 = 0$, we consider prior data a mixture of 25 Gaussian distributions, the same as considered in [13]. The Gaussian distribution has mean $(8i, 8j, \cdots, 8i, 8j) \in \mathbb{R}^d$ for $(i, j) \in \{-2, -1, 0, 1, 2\}^2$ and unit variance. Each (unnormalized) mixture weight is independently drawn according to a $\chi^2$ distribution.

For the measurement model considered in Figure 3, we generate $A$ in the following way. We first sample a $d \times d$ matrix with each entry sampled from the standard normal and compute its SVD to get $U$ and $V$ for $A$. The singular value is given by $[1, \cdots, 1/20]$ where each component in between is independently sampled from $\mathrm{Unif}([1/20, 1])$ such that the condition number of $A$ is 20. The observation noise is then determined by SNR. For the measurement model considered in Table 1, the matrix $U$ and $V$ for the SVD form of $A$ is the same to the above. Each singular value in $S$ is independently sampled from $\mathrm{Unif}([0, 1])$, and $\sigma$ is sampled from $\mathrm{Unif}([0.2 \max S, \max S])$.

For all the experiments we run the boosted posterior from $T^* - 0.01$ such that the ODE solution $Q, b$ is well-defined. We use BlackJAX [12] to implement the No-U-turn sampler.

Besides results reported in Figure 3, we further test tilted transport when $d' < d$. In this setting, $\lambda_{\min}(Q_t)$ remains zero but the signal corresponding to the non-zero eigenvalues still gets enhanced. Therefore, although it becomes more difficult for $v_{T^*}$ to be strongly log-concave, the tilted transport can still make the new posterior easier to sample even if it is not strongly log-concave yet. As shown in Table 1, when $d' = 0.9d$, 10% percent eigenvalues of $Q_t$ are zero, our tilted transport technique still reduces the statistical distance of the posterior samples significantly. We also consider an even more challenging case where $d' = 1$ such that the target posterior is still heavlily multimodal (as visualized in the 2D example in Figure 2). In this case, LMC suffers from the local maxima of the potential and thus cannot explore the multimodal distribution efficiently. We use the No-U-turn sampler[34], a Hamilton Monte Carlo (HMC) method, as the baseline method, which can move among different modes more efficiently than Langevin. We find that the tilted transport technique can still boost the performance of HMC in this challenging setting. We also verify Theorem 5 by sampling from the boosted posterior directly from its analytical formula and running the reverse SDE, and the obtained samples approximate the target posterior well, as reported in Table 1.

Table 1: Sliced Wasserstein distance for Gaussian mixture prior for degenerate case

| $d$ | $d' = 0.9d$ | | | $d' = 1$ | | |
|---|---|---|---|---|---|---|
| | Langevin | Boosted Langevin | Analytic Boost | HMC | Boosted HMC | Analytic Boost |
| 20 | $4.21 \pm 1.87$ | $\mathbf{2.32 \pm 2.42}$ | *$0.02 \pm 0.00$* | $1.33 \pm 1.02$ | $\mathbf{1.11 \pm 0.83}$ | *$0.12 \pm 0.07$* |
| 40 | $4.09 \pm 2.02$ | $\mathbf{2.45 \pm 1.79}$ | *$0.02 \pm 0.00$* | $2.04 \pm 1.26$ | $\mathbf{1.81 \pm 1.03}$ | *$0.13 \pm 0.07$* |
| 80 | $4.40 \pm 2.31$ | $\mathbf{2.75 \pm 2.10}$ | *$0.02 \pm 0.00$* | $2.98 \pm 2.15$ | $\mathbf{2.77 \pm 2.32}$ | *$0.11 \pm 0.06$* |

## E.2 Imaging Problems

We perform four inverse tasks on the Flickr-Faces-HQ Dataset (FFHQ) [39] to demonstrate the application of the tilted transport technique on imaging data as a proof of concept. To apply the proposed tilted transport technique to these problems, we still need to select a baseline method for sampling from the boosted posterior $\nu_{T^*}$. In the case of ill-conditioned problems, sampling $\nu_{T^*}$ may still be challenging for principled algorithms like LMC, and we still need to rely on heuristic methods for imaging tasks. However, as noted in the introduction, most existing heuristic methods primarily facilitate conditional generation based on the measurement, lacking principled guarantees for posterior sampling. Consequently, we lack a principled interpretation for enhancing these methods with tilted transport. Nevertheless, we can still experiment with such methods as a proof of concept. We chose Diffusion Model Based Posterior Sampling (DMPS) [48] as the baseline method for the following reasons: The main assumption of DMPS in approximating the time-dependent conditional score is that the prior $\pi$ is uninformative (flat) with respect to $X_t$, such that $p(X_0|X_t) \propto p(X_t|X_0)$. This assumption only holds approximately in early phases of the forward diffusion, and hopefully a higher SNR provided by tilted transport makes the effect of this approximation error smaller.

We conducted four tasks: (a) denoising; (b) inpainting with random masks from [53]; (c) 4× super-resolution; and (d) deblurring using a Gaussian kernel. Our algorithm was implemented using the NVIDIA codebase [47] with 1000 diffusion steps for posterior sampling, and utilized the score function from a pretrained diffusion model [20]. Similar to our Gaussian mixture model experiments where we adjusted the timing for the boosted posterior to avoid the singularity of $Q_t$, we shifted 6 - 10 timesteps for setting the boosted posterior. Experiments show that the final performance is robust with respect to the number of shifted steps. Figure 4 showcases examples from the inpainting task, demonstrating how tilted transport enhances the baseline DMPS method. Additionally, we report various sample statistics including peak signal-to-noise ratio (PSNR), structural similarity index (SSIM), and Learned Perceptual Patch Similarity (LPIPS). However, it is important to note that while these statistics assess the quality of prior data generation, they may not accurately reflect the quality of posterior samples.

Table 2: Performance for tasks on FFHQ Dataset.

| Task | Denoising | | Inpainting | | Super-resolution | | Deblur | |
|---|---|---|---|---|---|---|---|---|
| Metrics | DMPS | Boost | DMPS | Boost | DMPS | Boost | DMPS | Boost |
| PSNR(dB) ↑ | 32.153 | **32.350** | 22.458 | **23.312** | 26.761 | **26.899** | 29.088 | **29.155** |
| SSIM ↑ | 0.886 | 0.886 | 0.786 | **0.800** | 0.760 | **0.754** | 0.815 | 0.815 |
| LPIPS ↓ | 0.060 | **0.039** | 0.131 | **0.098** | 0.129 | **0.109** | 0.098 | **0.094** |

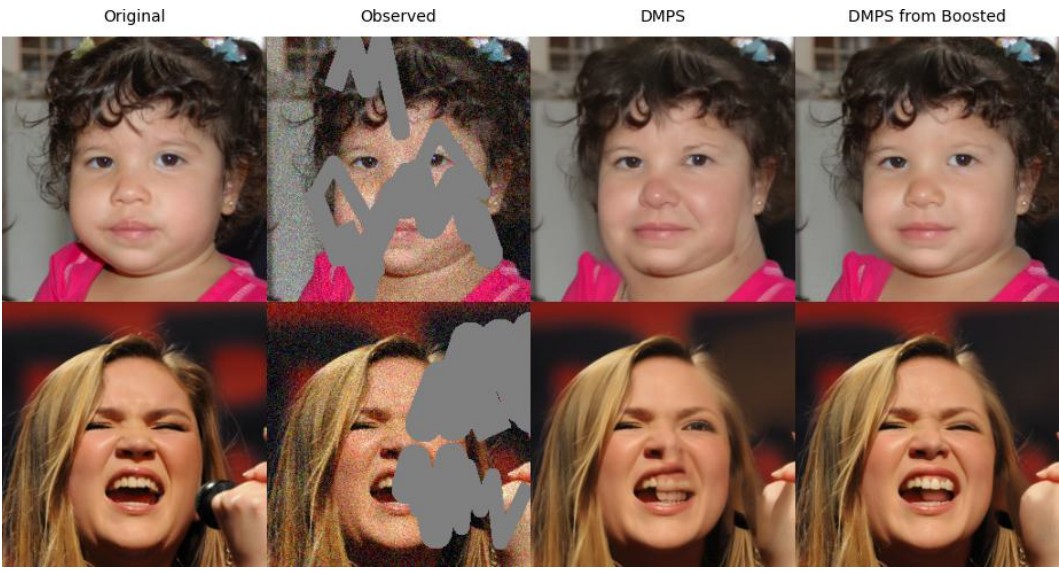

Figure 4: Examples for inpainting with random masks over FFHQ dataset

