# OpenReview forum: "Provable Posterior Sampling with Denoising Oracles via Tilted Transport"
_NeurIPS.cc/2024/Conference — NeurIPS 2024 poster_

### Official Review · Reviewer_SHHR · 2024-06-26

**Soundness:** 4
**Presentation:** 1
**Contribution:** 3
**Rating:** 6
**Confidence:** 3

**Summary:**

This work studies posterior sampling using unconditional diffusion models, or denoising oracles. The authors propose a "tilted transport" technique to combine a linear inverse problem and denoising oracles into a "boosted" posterior sampling problem. The authors study the feasibility of "boosted" posterior sampling with respect to properties (e.g. singular values) of $A$ (or $A^\top A$), the measurement operator. They devise conditions on these properties for which "boosted" posterior sampling is tractable.

**Strengths:**

* The subject (posterior sampling from denoising oracles) is interesting and relevant to the ML community.

**Weaknesses:**

* The presentation of this manuscript is very poor. First, the formulation is very different from other diffusion (or generative) models papers, with unusual notations and concepts, some of which are undefined in the text (e.g. quadratic tilt or $\ll$ at line 99). Second, it feels like the authors use this different formulation to purposely obfuscate their contribution. If I understand correctly, the "tilted transport" technique is an annealing of the likelihood $p(y | x_t)$ with respect to the added noise $\sigma_t^2 = 1 - \exp(-2t)$, such that it converges to $p(y | x)$ as $t$ goes to $0$. This is similar to previous works [1-4], but because the notations are so different, it is impossible for the reader to compare them. The authors should provide a thorough comparison of their method (preferably using standard formulation) with these previous works [1-4].

* In the introduction (lines 40-46), the error of previous works using heuristics (including [1-4]) is not "uncontrollable" but not analyzed. In fact, Rozet et al. [3] propose a sampling scheme that is exact for enough LMC (Langevin Monte Carlo) corrections steps. I don't see how this "contrasts with the principled sampling used in prior data generation".

* There is no algorithm to concisely describe the method. The provided code does not help as it is undocumented and frankly messy.

* The tilted transport technique is not applicable to any distribution $\pi$. In particular, sampling from $\nu_{T^*}$ is only possible in contrived settings. For most distributions $\pi$ and most measurement matrix $A$, sampling from $\nu_{T^*}$ is as hard as sampling from $\nu_0$.

* The experiments are not relevant for most general linear inverse problems. The prior distributions are low-dimensional ($d \leq 80$) mixture of few (25 components) Gaussians. The condition number of $A^\top A$ is controlled ($\kappa = 20$).

* In appendix, the authors compare their method with Meng et al. [5] which is a quite poor (and not peer-reviewed) baseline. Better baselines [1-4] are not considered.

[1] Song et al. "Pseudoinverse-Guided Diffusion Models for Inverse Problems". In International Conference on Learning Representations. 2023.

[2] Finzi et al. "User-defined Event Sampling and Uncertainty Quantification in Diffusion Models for Physical Dynamical System". In International Conference on Machine Learning. 2023.

[3] Rozet et al. "Score-based Data Assimilation". In Advances in Neural Information Processing Systems. 2023.

[4] Adam et al. "Posterior samples of source galaxies in strong gravitational lenses with score-based priors". 2022.

[5] Meng et al. "Diffusion model based posterior sampling for noisy linear inverse problems". 2022.

**Questions:**

* To my understanding, simulating the reverse SDE Eq. (3) from $T$ to $0$ only generates samples from $\pi_0$ if $\pi_t$ is the measure defined by applying the Ornstein-Uhlenbeck process to $\pi_0$, i.e. $\pi_t = C_{\beta_t} D_{\alpha_t} \pi_0$. Therefore, for $\nu_t = T_{Q_t, b_t} \pi_t$ to generate samples from $\nu_0 = T_{Q, b} \pi_0$ with the reverse SDE, it would be necessary that $\nu_t = C_{\beta_t} D_{\alpha_t} \nu_0$. Is this the case?

* What is the relationship of $\nu_t$ with previous heuristics [1-4] ?

### Typos

* Line 110, $w \sim \gamma_{d'}$
* Line 122, $Q = \sigma^{-2} I_d$
* Line 222, $p(\tilde{y} | x) = p(X_{T^*} | X_0)$

**Limitations:**

See weaknesses.

---

> ### Author Rebuttal · Authors · 2024-08-06
>
> Thank you for reviewing our submission. We regret that you have a rather negative view of the paper. It seems to us that there are some fundamental misunderstandings of our tilted transport technique. We hope that our clarifications below, as well as the other reviews, can help resolve these misunderstandings. Below we address specific questions. The response is split due to length constraints.
>
> > The presentation of this manuscript is very poor. First, the formulation is very different from other diffusion models papers, with unusual notations and concepts, some of which are undefined in the text (e.g. $\ll$ at line 99). Second, it feels like the authors use this different formulation to purposely obfuscate their contribution. If I understand correctly, the "tilted transport" technique is an annealing of the likelihood $p(y|x_t)$ with respect to the added noise $\sigma_t^2 = 1-\exp(-2t)$, such that it converges to $p(y|x)$ as $t$ goes to 0. This is similar to previous works [1-4], but because the notations are so different, it is impossible for the reader to compare them. The authors should provide comparison of their method with these previous works].
>
> We would like to first clarify that our formulation of the score-based diffusion in terms of the Ornstein–Uhlenbeck (OU) forward process is quite standard and is also known as variance preserving. As further explained in the footnote, the commonly used noise schedule $dX_t = -\beta(t)X_t~dt + \sqrt{2\beta(t)}dW_t$ can be interpreted as a time-rescaled version of the standard OU process, ensuring that all our results directly apply.
>
> Second, we clarify that the "tilted transport technique" is not equivalent to annealing the likelihood, if by "annealing" the reviewer refers to a process that adjusts noise levels or, equivalently, temperature. In contrast, our technique operates based on the following two facts that we prove:
>
> 1. We can construct a time-dependent family of likelihoods $\nu_t \propto \exp\left(-\frac{1}{2} x^\top Q_t x + x^\top b_t\right)$ such that at $t=0$, this likelihood matches our desired posterior.
>
> 2. The same reversed SDE dynamics used in prior data generation can transport this time-dependent likelihood from $T^*$ to 0 exactly, and we can prove that $\nu_{T^*}$ is much easier to sample than $\nu_0$.
>
> The second fact above also highlights the advantages of our method: if we ignore time discretization error, a single step of the reverse SDE over $\delta t$ directly moves the sample from $\nu_{t+\delta t}$ to $\nu_t$ in the distributional sense. In contrast, if we have two distributions with noise levels $\sigma_{t+\delta t}$ and $\sigma_t$, we can only ensure that these two distributions are close, but moving from the former to the latter may require multiple steps of Langevin Monte Carlo, which is much less efficient. Further discussion on this point is provided in our reply to the third point.
>
> Finally, we would like to clarify that for two measures $\nu$ and $\mu$, the notation $\nu \ll \mu$ is a standard notation in measure theory, indicating that $\nu$ is absolutely continuous with respect to $\mu$, meaning the Radon–Nikodym derivative $\frac{d\nu}{d\mu}$ exists. More details can be found at Radon–Nikodym theorem.
>
> > What is the relationship of $\nu_t$ with previous heuristics [1-4]?
>
> Our $\nu_t$ is fundamentally different from previous heuristics [1-4]. We have established that if an accurate score is obtained from training on prior data, one can (1) first sample $\nu_{T^*}$ and (2) run the reverse dynamics from $T^*$ to 0 to get samples from $\nu_0$. We provide sufficient conditions under which $\nu_{T^*}$ allows for fast sampling, ensuring that the total cost of both steps is low while the final result is accurate.
>
> In contrast, heuristics [1-4] encounter significant approximation errors. They assume that the probability $p(x_t | x_0)$ follows a Gaussian distribution with its mean given by Tweedie’s formula. This assumption is generally not true, even for a simple Gaussian mixture prior. Without LMC corrections, achieving accurate sampling with these methods requires extreme conditions. If LMC corrections are used, it is difficult to estimate in advance how many correction steps are needed to guarantee accurate sampling. Our method, therefore, offers a more principled and efficient approach with clear conditions for achieving accurate results.
>
>
> > In the introduction (lines 40-46), the error of previous works using heuristics (including [1-4]) is not "uncontrollable" but not analyzed. In fact, Rozet et al. [3] propose a sampling scheme that is exact for enough LMC (Langevin Monte Carlo) corrections steps. I don't see how this "contrasts with the principled sampling used in prior data generation".
>
> The difference between our method and other heuristics has been clarified above. We agree with the reviewer that if one uses enough LMC correction steps, the algorithm in Rozet et al. can be nearly exact. In fact, for any sampling algorithm, augmenting it with enough LMC steps will result in nearly exact samples — even running LMC directly on the posterior likelihood.
> However, we believe such a guarantee is insufficient since the required number of LMC correction steps can be unfeasibly large without isoperimetry assumptions. For instance, sampling a Gaussian mixture using LMC may take an exponential number of steps in relation to the dimension.
>
> In contrast, the prior data generation in score-based diffusion is substantially more powerful, as it comes with quantitative error bounds for every source, including initialization error, time discretization error, and score estimation error. The goal of our paper is to achieve a similar level of quantitative control for posterior sampling. Specifically, we provide explicit conditions ensuring that $\nu_{T^*}$ is log-concave, allowing LMC to sample it with fast convergence, and the remaining cost associated with tilted transport is similar to prior data generation.

---

> ### Author Response · Authors · 2024-08-06
> **Rebuttal by Authors (cont.)**
>
> > There is no algorithm to concisely describe the method. The provided code does not help as it is undocumented and frankly messy.
>
>
> We provide a description of the algorithm in the PDF for the global response. We will improve the readability of the code in a later revision.
>
>
> > The tilted transport technique is not applicable to any distribution $\pi$. In particular, sampling from $\nu_{T^*}$ is only possible in contrived settings. For most distributions $\pi$ and most measurement matrix $A$, sampling from $\nu_{T^*}$ is as hard as sampling from $\nu_0$.
>
> While the reviewer is technically correct that our tilted transport technique is not applicable for arbitrary $\pi$, we note that in Section 3 we show that such universal guarantees are impossible. Our Corollary 10 then lays out sufficient conditions that ensure efficient sampling. We are not sure to understand the second remark, since Figure 2 in the paper shows a phase diagram indicating the interplay between A and pi that fullfils our sufficient guarantees. We also note that in Appendix C we describe an iterated tilted transport that operates under more general conditions, at the expense of introducing additional sampling error (but which can be quantified).
>
> > The experiments are not relevant for most general linear inverse problems. The prior distributions are low-dimensional ($d\leq 80$) mixture of few (25 components) Gaussians. The condition number of $A^\top A$ is controlled ($\kappa=20$).
>
> We shall remark that a majority of existing methods cannot sample the posterior in this case, as noted in [1,2]. Our method, however, has both theoretical guarantees and numerical support for accurately sampling the posterior when the condition number is not too large. This is demonstrated in our numerical experiments, which validate our theoretical claims.
>
> [1] Cardoso et al. Monte Carlo guided Denoising Diffusion models for Bayesian linear inverse problems. (2024)
>
> [2] Janati et al. Divide-and-Conquer Posterior Sampling for Denoising Diffusion Priors (2024)
>
>
> > In appendix, the authors compare their method with Meng et al. [5] which is a quite poor (and not peer-reviewed) baseline. Better baselines [1-4] are not considered.
>
> Thank you for your comments. Our main paper primarily focuses on a provable and principled posterior sampling algorithm, and we numerically demonstrate the algorithm on Gaussian mixture priors where we can precisely characterize the ground-truth posterior and quantitatively compute the statistical quality of the obtained samples. In contrast, our experiments on imaging are preliminary. Although we demonstrated that the boosted posterior $\nu_{T^*}$ is easier to sample than $\nu_0$, $\nu_{T^*}$ may still present significant sampling challenges, leading to non-negligible errors. Consequently, a deeper investigation is required to understand how the tilted transport technique performs when $\nu_{T^*}$ has such errors.
>
> Furthermore, it is much more difficult to assess sample quality in the posterior distribution sense for imaging problems, as a single realistic image does not necessarily imply more accurate posterior sampling. Therefore, we have decided to focus the main text on the theoretical aspects and include the imaging experiments in the appendix as potential future directions. We acknowledge the need for comparisons with better baselines and will consider these in future work as we further develop and test our method.
>
>
> > To my understanding, simulating the reverse SDE Eq. (3) from T to 0 only generates samples from $\pi_0$ if $\pi_t$ is the measure defined by applying the Ornstein-Uhlenbeck process to $\pi_0$, i.e. $\pi_t = C_{\beta_t}D_{\alpha_t}\pi_0$. Therefore, for $\nu_t = T_{Q_t, b_t} \pi_t$ to generate samples from $\nu_0$ with the reverse SDE, it would be necessary that $\nu_t = C_{\beta_t}D_{\alpha_t}\pi_0$. Is this the case?
>
> That is not the case. To be more specific, your first sentence is correct. However, the key finding of our tilted transport technique is that for $\nu_t = T_{Q_t, b_t} \pi_t$, which is NOT in the form of $\nu_t = C_{\beta_t}D_{\alpha_t}\pi_0$, the reverse SDE Eq. (3) from $t$ to $0$ can generate samples EXACTLY without any approximation from the target posterior $\nu_0 = T_{Q_0, b_0} \pi_0$.
>
>
> > Typos.
>
> Thanks for pointing out these typos.

---

> ### Comment · Reviewer_SHHR · 2024-08-07
> **Answer by Reviewer**
>
> Thank you for your detailed rebuttal and taking the time to answer my questions. Upon reading the algorithm provided in the global rebuttal, I notice that you are using $\nabla \log \pi_t(x)$ to simulate the reverse SDE and not $\nabla \log \nu_t(x)$ like previous posterior sampling methods would. Am I correct in understanding that $\nu_t(x)$ is only used to sample the initial $X_{T^*}$ then? If it is the case, the proposed method is indeed fundamentally different from previous methods and I am sorry for the rather harsh review.

---

> > ### Author Response · Authors · 2024-08-07
> >
> > Thank you for your quick response. Your updated understanding is correct: we construct the time-dependent family $\nu_t(x)$ and prove that the same reverse dynamics based on $\nabla \log \pi_t(x)$ used for prior data generation can transport samples along this family of distributions. This explanation is for theoretical understanding. At the algorithm level, we do only need $\nu_{T^*}$ to sample $X_{T^*}$ (or $\nu_{T^*-\epsilon}$ to sample $X_{T^*-\epsilon}$ for numerical stability).
> >
> > Given your updated understanding of our approach, we hope you might consider revisiting the score for our submission. We appreciate your review and your willingness to understand our work in more detail.

---

> ### Comment · Reviewer_SHHR · 2024-08-07
>
> Thank you, it makes much more sense. Based on my updated understanding, I now believe that this work has potential for high impact. However, I agree with reviewer **jPCg** that the manuscript lacks accessibility, which I partly attribute my misunderstanding to. Therefore, even though the contribution seems clearer for me now, you should consider making it more accessible.
>
> In this light, I now argue for acceptance and will raise my score to 6 (previously 3).

---

### Official Review · Reviewer_jPCg · 2024-07-11

**Soundness:** 3
**Presentation:** 3
**Contribution:** 3
**Rating:** 6
**Confidence:** 4

**Summary:**

The paper presents a theory regarding posterior sampling in linear inverse problems with additive Gaussian noise. It provides two major contributions:

* A proof that no general algorithm exists for posterior sampling of Ising models when the degradation operator is ill-conditioned.
* A transport equation for the coefficients of the quadratic form of the posterior distribution in the linear setup, transforming it into a new distribution that is easier to sample from. This includes conditions on SNR and condition number for strong log-concavity of the transformed probability, allowing for efficient sampling.

The authors illustrate their theoretical results with a toy example with Gaussian mixture model.

**Strengths:**

* The paper effectively presents the problem and necessary background, along with detailed and rigorous derivations.
* In the context of the linear setup, the paper presents impactful and theoretically sound results regarding posterior sampling.

**Weaknesses:**

* Figure 1 and its description are not clear. I would expect it to clearly summarize the essence of the work, particularly highlighting the second contribution.

* While the paper contains rigorous derivations, it lacks intuitive explanations and simplified examples, making it difficult for readers to understand the overall direction and implications of the work as they read.
This lack of accessibility limits the paper's potential audience.

* The focus on linear problems with additive Gaussian noise limits the practicality and potential impact of the work.

* The lack of a practical algorithm for implementing the tilted transport map limits the applicability of this work.

* The authors' decision to place the extension to nonlinear setups and imaging experiments to the appendix is puzzling, as these results are compelling and could strengthen the paper's main arguments if presented in the main body.

**Questions:**

Please address weaknesses.

**Limitations:**

The limitations of the work, e.g. the focus on linear setups, are only briefly mentioned in the discussion section. This warrants a more extensive and dedicated discussion to adequately address their implications.

---

> ### Author Rebuttal · Authors · 2024-08-06
>
> Thank you for your time and effort in reviewing our submission. We will do our best below to address your concerns and queries.
>
> > Figure 1 and its description are not clear. I would expect it to clearly summarize the essence of the work, particularly highlighting the second contribution.
>
> Our technique is based on the following two proven facts:
>
> 1. We can construct a time-dependent family of likelihoods $\nu_t \propto \exp\left(-\frac{1}{2} x^\top Q_t x + x^\top b_t\right)$ such that at $t=0$, this likelihood matches our desired posterior.
>
> 2. The same reversed SDE dynamics used in prior data generation can transport this time-dependent likelihood from $T^*$ to 0 exactly. We have proven that $\nu_{T^*}$ is much easier to sample than $\nu_0$.
>
> Figure 1 is intended to demonstrate our technique in the Gaussian mixture case, where the bottom row represents our technique and the top row represents the standard prior data generation. We will revise the description of Figure 1 to better explain this.
>
> The second fact highlights the advantages of our method: if we ignore time discretization error, a single step of the reverse SDE over $\delta t$ directly moves the sample from $\nu_{t+\delta t}$ to $\nu_t$ in the distributional sense. In contrast, with two distributions having noise levels $\sigma_{t+\delta t}$ and $\sigma_t$, we can only ensure that these distributions are close, but moving from one to the other may require multiple steps of Langevin Monte Carlo, which is much less efficient.
>
>
>
> > While the paper contains rigorous derivations, it lacks intuitive explanations and simplified examples, making it difficult for readers to understand the overall direction and implications of the work as they read. This lack of accessibility limits the paper's potential audience.
>
> Thank you for your comments. We intended to use the paragraph "A Motivating Example" at the beginning of Section 4 as a simplified example to provide readers with some intuitive explanations. For this example, most methods in the literature cannot sample the posterior exactly, while our tilted transport technique can solve this task elegantly as a special case. We will better illustrate this intuition using this example in the revision.
>
> > The focus on linear problems with additive Gaussian noise limits the practicality and potential impact of the work.
>
> While our primary focus is on linear problems with additive Gaussian noise, it's worth noting that additive Poisson noise, another common type of noise in the literature, can be well approximated by Gaussian noise with high accuracy; see [1]. Thus, our method is also applicable to Poisson noise. We acknowledge that our results are heavily dependent on the linear structure of the problem, which allows us to derive a principled and provable posterior sampling algorithm. To the best of our knowledge, there is currently no principled algorithm for nonlinear problems, even when using a Gaussian mixture as the prior. The only exceptions might be methods based on sequential Monte Carlo, which guarantee asymptotic performance but may require an impractically large number of particles. Therefore, we believe that developing a principled algorithm for nonlinear problems remains an open challenge for future research.
>
> [1] Chung et al. “Diffusion Posterior Sampling for General Noisy Inverse Problems”, ICLR 2023.
>
> > The lack of a practical algorithm for implementing the tilted transport map limits the applicability of this work.
>
> We provide a description of the algorithm in the PDF for the global response.
>
> > The authors' decision to place the extension to nonlinear setups and imaging experiments to the appendix is puzzling, as these results are compelling and could strengthen the paper's main arguments if presented in the main body.
>
> We would like to clarify that our appendix does not address any nonlinear setups. The iterated tilted transport discussed in Appendix C is still about linear problems but with even larger condition numbers, in which the boosted posterior $\nu_{T^*}$ may still be challenging to sample. While this idea is promising, it is still relatively immature compared to the one-round tilted transport presented in the main text. Therefore, we believe it is more appropriate to include it in the appendix to highlight a potential future direction without causing unnecessary confusion.
>
> We appreciate the positive evaluation of our imaging experiments. These experiments are promising but also preliminary. Our main paper primarily focuses on *provable and principled* posterior sampling based on score-based diffusion models. For imaging problems, although we demonstrated that the boosted posterior $\nu_{T^*}$ is easier to sample than $\nu_0$, $\nu_{T^*}$ may still present significant sampling challenges, leading to non-negligible errors. Consequently, a deeper investigation is required to understand how the tilted transport technique performs when $\nu_{T^*}$ has such errors. Therefore, we have decided to focus the main text on the theoretical aspects and include the imaging experiments in the appendix as potential future directions.

---

> > ### Comment · Reviewer_jPCg · 2024-08-11
> >
> > I appreciate the clarifications provided in the rebuttal.
> > I believe my current score is appropriate.

---

### Official Review · Reviewer_G6DH · 2024-07-12

**Soundness:** 4
**Presentation:** 4
**Contribution:** 4
**Rating:** 8
**Confidence:** 4

**Summary:**

This paper proposes a novel posterior sampling algorithm specialized to linear inverse problems. The proposed algorithm relies on a sequence of intermediate and **tractable** posterior distributions $\nu_t$ that satisfy a transport equation up to some time $T^*$. The algorithm then operates as follows; first draw approximate samples from the distribution $\vu_{T^*}$ using any approximate inference algorithm, and then run the reverse SDE from $T^*$ to $0$ starting at the obtained samples. Of course, the computational hardness is transferred to $\nu_{T^*}$ and the authors derive sufficient conditions that ensure it is strongly log concave. The study of the algorithm is then specialized to both the Ising model and Gaussian mixtures. They show for example that $\nu_{T^*}$ is strongly log-concave for the Ising model under the condition that the eigenvalues difference is smaller than 1, thus providing a new efficient continuous time procedure for sampling from it.

**Strengths:**

Overall, this paper is highly original and I greatly appreciated reading it. It has quite a few contributions and the proposed main algorithm is interesting. The writing is clean so that the paper is relatively easy to understand, while still sufficiently detailed. The properties of the algorithm are well investigated throughout the Ising model and Gaussian mixture examples.

**Weaknesses:**

There isn’t any serious weakness; perhaps I would have liked to see more challenging experiments and comparisons with denoising diffusion based posterior sampling algorithms, but in my opinion this is optional for this paper, since it is more on the theoretical side and has strong contributions.

**Questions:**

N/A

---

> ### Author Rebuttal · Authors · 2024-08-06
>
> We thank the reviewer very much for the time and effort spent in reviewing our paper, and for the positive evaluation.

---

### Official Review · Reviewer_ue87 · 2024-07-14

**Soundness:** 4
**Presentation:** 4
**Contribution:** 4
**Rating:** 8
**Confidence:** 1

**Summary:**

The paper "Provable Posterior Sampling with Denoising Oracles through Tilted Transport" analyses the possibility of posterior sampling utilizing denoising oracles, such as score-based diffusion models. To demonstrate the possibilities the authors focus on linear inverse problems and propose the tilted transport technique. It uses the quadratic log-likelihood structure of linear inverse problems together with the oracle to transform the original posterior into one easier to sample. Based on the example of the Ising model the authors show that well-conditioned measurements allow the transport to simplify the task. Furthermore, first empirical tests for image tasks indicate promising practical use.

**Strengths:**

Well written and well structured with proofs. Potential for impact is high

**Weaknesses:**

The topic is far from my expertise. I cannot see obvious weaknesses for this analysis. It has its limitations, but it is clearly stated and argued by the authors.

**Questions:**

What is the reason for the peak with large percentiles for 10^-3 SNR in Figure 3?

**Limitations:**

yes

---

> ### Author Rebuttal · Authors · 2024-08-06
>
> We thank the reviewer very much for the time and effort spent in reviewing our paper, and for the positive evaluation.  Below we address your specific question.
>
> > What is the reason for the peak with large percentiles for 10^-3 SNR in Figure 3?
>
> The peak observed with large percentiles for an SNR of $10^{-3}$ in Figure 3 is precisely predicted by our theoretical analysis in Corollary 10. The text following Corollary 10 and Figure 2 provides a detailed interpretation of this result. Specifically, given a prior and $\kappa(A)$, the target posterior can be reliably sampled if the SNR is either sufficiently low or high, as the sufficient condition in Corollary 10 is met. However, for moderate SNR values, the sufficient condition in Corollary 10 may not be satisfied. Consequently, while the transported measure $\nu_{T^*}$ may be easier to sample than $\nu_0$, it may still pose challenges for Langevin Monte Carlo sampling. As a result, the final statistical error between our samples and the target posterior might be larger for moderate SNRs compared to extremely low or high SNRs. The peak observed for an SNR of $10^{-3}$ in Figure 3 provides strong numerical verification of our theoretical predictions.

---

> > ### Comment · Reviewer_ue87 · 2024-08-13
> >
> > Thank you for the clarification

---

### Author Rebuttal · Authors · 2024-08-06

We would like to thank all reviewers for their time and effort in reviewing our paper and for providing valuable suggestions that imrpove the draft. We address most points raised by individual reviewers in separate rebuttals. **Here we attach a line-by-line description of our tilted transport algorithm in the PDF to clarify the confusion of the algorithm from two reviewers.**

To further clarify, in our paper, we provide explicit conditions ensuring that the boosted posterior $\nu_{T^*}$ is log-concave, which allows Langevin Monte Carlo (LMC) to sample it with fast convergence. The cost associated with tilted transport is comparable to prior data generation. This feature stands in contrast to most existing methods that rely on LMC or particle methods to guarantee accuracy. In those methods, the number of LMC steps or particles required is hard to estimate a priori and can potentially grow exponentially with the dimension.

---

### Decision · Program_Chairs · 2024-09-25

**Decision:**

Accept (poster)

**Comment:**

The paper presents a method for posterior sampling using a tilted transport method and focusing on linear inverse problems. Four reviewers all agreed that the paper was above threshold for acceptance and not a borderline case. While one strong accept was very low confidence and both strong accepts were light in content, the consensus indicates that the paper should be accepted. One reviewer who initially had concerns was convinced to change their view during the feedback period interaction.  The paper is technically interesting and novel and should be accepted.